# Quadruple gene-engineered natural killer cells enable multi-antigen targeting for durable antitumor activity against multiple myeloma

Frank Cichocki [1,6], Ryan Bjordahl[2,6], Jodie P. Goodridge[2], Sajid Mahmood[2], Svetlana Gaidarova[2], Ramzey Abujarour[2], Zachary B. Davis[1], Aimee Merino [1], Katie Tuininga[1], Hongbo Wang[1], Akhilesh Kumar[1], Brian Groff[2], Alec Witty[2], Greg Bonello[2], Janel Huffman[2], Thomas Dailey[2], Tom T. Lee[2], Karl-Johan Malmberg [3], Bruce Walcheck[4], Uta Höpken [5], Armin Rehm [5], Bahram Valamehr [2] ✉ & Jeffrey S. Miller [1] ✉

Allogeneic natural killer (NK) cell adoptive transfer is a promising treatment for several cancers but is less effective for the treatment of multiple myeloma. In this study, we report on quadruple gene-engineered induced pluripotent stem cell (iPSC)-derived NK cells designed for mass production from a renewable source and for dual targeting against multiple myeloma through the introduction of an NK cell-optimized chimeric antigen receptor (CAR) specific for B cell maturation antigen (BCMA) and a high affinity, non-cleavable CD16 to augment antibody-dependent cellular cytotoxicity when combined with therapeutic anti-CD38 antibodies. Additionally, these cells express a membrane-bound interleukin-15 fusion molecule to enhance function and persistence along with knock out of *CD38* to prevent antibody-mediated fratricide and enhance NK cell metabolic fitness. In various preclinical models, including xenogeneic adoptive transfer models, quadruple gene-engineered NK cells consistently demonstrate durable antitumor activity independent of exogenous cytokine support. Results presented here support clinical translation of this off-the-shelf strategy for effective treatment of multiple myeloma.

Multiple myeloma (MM) is a clonal plasma cell neoplasm that constitutes ~10% of hematologic malignant disorders[1]. Because of the high involvement of MM in the bone marrow, most patients present with anemia and osteolytic skeletal lesions that cause fatigue and bone pain[2]. Advances in the treatment of MM with drugs such as thalidomide, bortezomib, and lenalidomide have extended overall patient survival[3–5]. Recently, monoclonal antibodies have become an essential part of the therapeutic arsenal in combination with other immunomodulatory drugs[6]. The anti-CD38 monoclonal IgGκ antibody daratumumab has direct tumor targeting and immunomodulatory mechanisms of action[7–10]. Daratumumab is approved as monotherapy and in combination with existing treatment regimens for patients with

[1]University of Minnesota, Department of Medicine, Minneapolis, MN 55455, USA. [2]Fate Therapeutics, San Diego, CA 92121, USA. [3]Oslo University Hospital, Oslo, Norway. [4]University of Minnesota, Department of Veterinary and Biomedical Sciences, St. Paul, MN 55108, USA. [5]Max-Delbrück-Center for Molecular Medicine, MDC, Berlin, Germany. [6]These authors contributed equally: Frank Cichocki, Ryan Bjordahl. ✉e-mail: bob.valamehr@fatetherapeutics.com; mille011@umn.edu

newly diagnosed MM or relapsed/refractory disease. In a phase 3 study, daratumumab in combination with lenalidomide and dexamethasone was associated with longer progression-free survival for patients with relapsed/refractory MM compared to treatment with lenalidomide and dexamethasone alone[11]. While advances have been made over the past 2 decades, the approach to the treatment of relapsed MM is complicated, and the disease continues to be incurable.

Immunotherapy is a relatively new but promising approach to the treatment of MM. Several chimeric antigen receptor (CAR) T-cell products have been created, and many of them target B-cell maturation factor antigen (BCMA), a surface molecule present in normal B lymphocytes and malignant plasma cells. Overall response rates in clinical trials testing anti-BCMA CAR-T cells have been impressive with complete response rates ranging from 27% to 55%[12–17]. However, manufacturing complexity resulting in high cost and product variability, BCMA antigen escape, and challenging toxicity profiles have hindered the full potential of anti-BCMA CAR cell therapies. To the latter point, cytokine release syndrome (CRS) is the most frequent serious acute anti-BCMA CAR-T cell-related toxicity, with an incidence of ~90% in most trials. Neurotoxicity is also common, with an incidence between 18% and 32%. Off-target toxicities, including cytopenia, also occur in most patients receiving anti-BCMA CAR-T cell therapy[18]. Regarding antigen loss, while the initial results from anti-BCMA CAR-T cell trials have been encouraging, biallelic deletion of the *BCMA* locus has been described as a mechanism of BCMA loss in MM patients treated with CAR-T cells[19,20]. BCMA can also be shed from the surface of plasma cells due to cleavage by γ-secretase, attenuating the efficacy of anti-BCMA CAR-T cell treatment[21]. Thus, additional immunotherapeutic approaches are needed to address MM heterogeneity and antigen escape.

Natural killer (NK) cells represent an attractive alternative as an adoptive cell therapy product. They are safe and effective in the allogeneic setting, and NK cell adoptive transfer is not associated with significant CRS or neurotoxicity[22,23]. However, low numbers of NK cells circulating in peripheral blood, donor variability, and impediments to genetic manipulation make scalable manufacture of uniformly edited NK cells with the potential for multi-dose administration a challenge. To overcome these limitations, we developed a platform for large-scale expansion of uniformly engineered NK cells derived from clonally expanded induced pluripotent stem cells (iPSCs). The engineered clonal iPSC lines are banked and used as a renewable source for off-the-shelf availability of engineered NK cells[24]. We refer to these cells as "iNK" (iPSC-derived NK) cells. Previously, we've shown that iNK cells have broad cytotoxic activity, efficiently recruit T cells, and augment checkpoint blockade therapy[24].

Here, we describe a uniform population of quadruple gene-modified iNK cells termed "iDuo-MM" CAR-NK cells tailored for the treatment of MM. To maximize antibody-dependent cellular cytotoxicity (ADCC), iDuo-MM CAR-NK cells were engineered to express a high affinity, non-cleavable version of the CD16a (hnCD16) Fc receptor containing the high-affinity polymorphism shown to improve therapeutic antibody efficacy[25] and engineered resistance to activation-induced cleavage by the ADAM17 metalloprotease[26]. The second edit was the introduction of an anti-BCMA CAR optimized for NK cell signaling with a high-avidity BCMA binder shown to enhance BCMA targeting[27]. The last two edits were the introduction of a membrane-bound IL-15/IL-15R fusion (IL-15RF) molecule to promote survival in the absence of exogenous cytokines and *CD38* knockout to prevent anti-CD38 mAb-mediated fratricide and enhance metabolic fitness of NK cells[28]. We demonstrate the ability of iDuo-MM CAR-NK cells to effectively engage in both anti-BCMA CAR-mediated cytotoxicity and hnCD16-mediated ADCC in combination with daratumumab. Additionally, we show that the combination of enhanced antitumor activity and augmented persistence results in durable in vivo control of MM by iDuo-MM CAR-NK cells that can be further enhanced through the administration of daratumumab alone or in combination with

γ-secretase inhibition to optimize dual-antigen targeting. These successful preclinical data support the clinical translation of iDuo-MM CAR-NK cells for the treatment of MM.

## Results

### Selection and functional validation of iDuo-MM CAR-NK cells

Engineered iPSC candidate clones for the derivation of CAR-iNK cells were generated by introducing an anti-BCMA CAR into iPSCs carrying the backbone edits of hnCD16, IL-15RF, and *CD38* knockout shown to promote the preferred adaptive state of NK cells[28]. Various CAR designs were screened, with the anti-BCMA-CAR4 construct containing intracellular signaling modalities derived from NKG2D, 2B4, and CD3ζ ultimately selected as the best-performing CAR design[29]. The selected iPSC candidate clones were initially assessed for differentiation capacity, NK cell lineage commitment, and uniformity of transgene expression. Phenotypically, iDuo-MM CAR-NK cells were uniformly positive for CD56, CD16, IL-15RF, anti-BMCA-CAR, DNAM-1, NKp30, and NKp46, and uniformly negative for CD38. The levels of KIR2DL1 and KIR2DL2/3 on iDuo-MM CAR-NK cells were low (Fig. 1a). In flow cytometry analyses, both unmodified iNK cells and iDuo-MM CAR-NK cells uniformly differentiated along the NK cell lineage. Compared to unmodified iNK cells, iDuo-MM CAR-NK cells had high levels of hnCD16 and IL-15RF, lacked CD38, and expressed high levels of the anti-BCMA CAR (Fig. 1b).

To rigorously test the cytotoxic potency and persistence of engineered iNK cells at low effector-to-target (E:T) ratios, we developed an in vitro serial restimulation assay. MM.1S cells transduced with NucLight Red were plated on the day of assay initiation followed by the addition of anti-BCMA CAR-iNK cells or backbone iNK cells at a 1:1 E:T ratio in the absence or presence of daratumumab. After 48 hours of co-culture (round 1), non-adherent effector cells were transferred to freshly plated MM.1S cells (round 2) without E:T ratio recalibration. This process was repeated (round 3), and MM.1S target cell survival was assessed in real-time with IncuCyte imaging. Notably, these assays were performed without the addition of exogenous cytokine. As anticipated, iNK cells derived from clones lacking an anti-BCMA CAR did not lyse MM.1S cells, which are known to be resistant to natural cytotoxicity[28]. The addition of daratumumab to backbone iNK cell co-cultures resulted in MM.1S cell death via ADCC, albeit with variability in round 3 of the assay. Importantly, anti-BCMA CAR-iNK cells displayed robust and sustained cytotoxicity against MM.1S cells alone and in combination with daratumumab in all three rounds of the assay (Fig. 2). Based on these results, anti-BCMA CAR-iNK cells were carried forward for subsequent in vitro and in vivo experiments.

### iDuo-MM CAR-NK cells exhibit antigen-dependent cytotoxicity

To determine antigen-specific function, we initially tested independent batches of iDuo-MM CAR-NK cells alongside iNK cells derived from two different backbone clones in a 4-hour cytotoxicity assay using wild-type Nalm6 cells that lack BCMA and Nalm6 cells with transgenic expression of BCMA as targets. As expected, iNK cells derived from the two backbone clones and iDuo-MM CAR-NK cells exhibited innate cytotoxic responses against wild-type Nalm6 cells lacking BCMA, which was measurable but weak in both cases (Fig. 3a, left). In contrast, we observed strong, dose-dependent specific cytotoxicity by iDuo-MM CAR-NK cells against BCMA-engineered Nalm6 cells (Fig. 3a, right). To confirm that iDuo-MM CAR-NK cells maintain tolerance towards healthy lymphocytes, we performed mixed co-culture assays where cytotoxicity against MM.1S cells and healthy allogeneic peripheral blood mononuclear cells (PBMCs) were assessed concurrently. For comparison, parallel assays were performed with PBNK cells activated overnight with IL-15. As anticipated, iDuo-MM CAR-NK cells exhibited higher levels of cytotoxicity against MM.1S cells compared to primed PBNK cells. While antigen-specific cytotoxicity was potent, off-target cytotoxicity against healthy allogeneic PBMCs was low for iDuo-MM

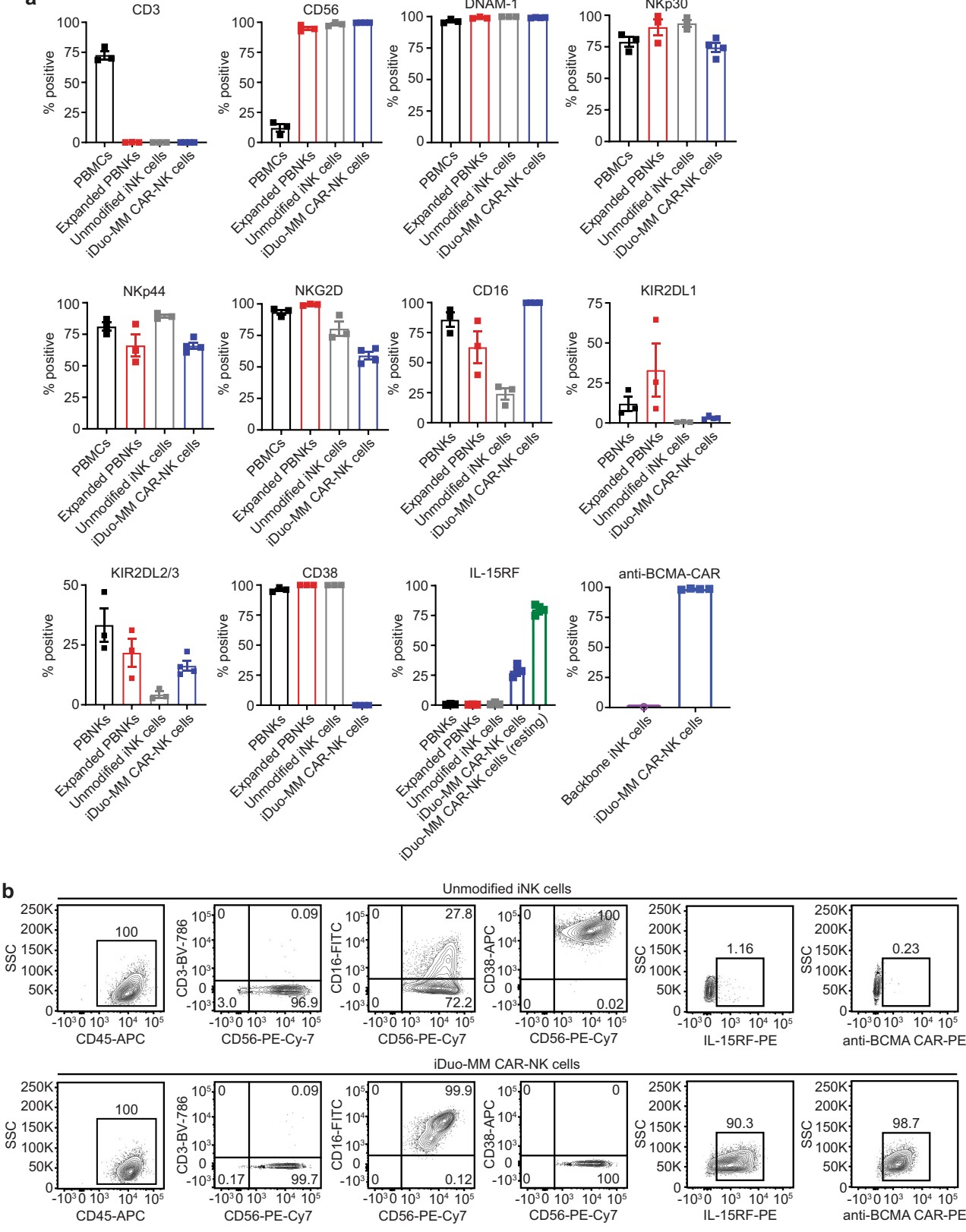

**Fig. 1 | Phenotype of iDuo-MM CAR-NK cells. a** Flow cytometry analysis of the indicated cell-surface proteins on PBMCs and PBNK cells from three healthy donors, unmodified iNK cells (*n* = 3 batches), and iDuo-MM CAR-NK cells (*n* = 4 batches). Backbone iNK cells (*n* = 3 batches) were also stained for comparison of anti-BCMA CAR levels. Data are presented as mean values ± SD. **b** Flow cytometry analysis of CD56 and cell-surface anti-BCMA CAR levels on iNK cells differentiated from an iPSC clone with backbone edits only, an iPSC clone with backbone edits, and the anti-BCMA-CAR4 construct (iDuo-MM CAR-NK cells).

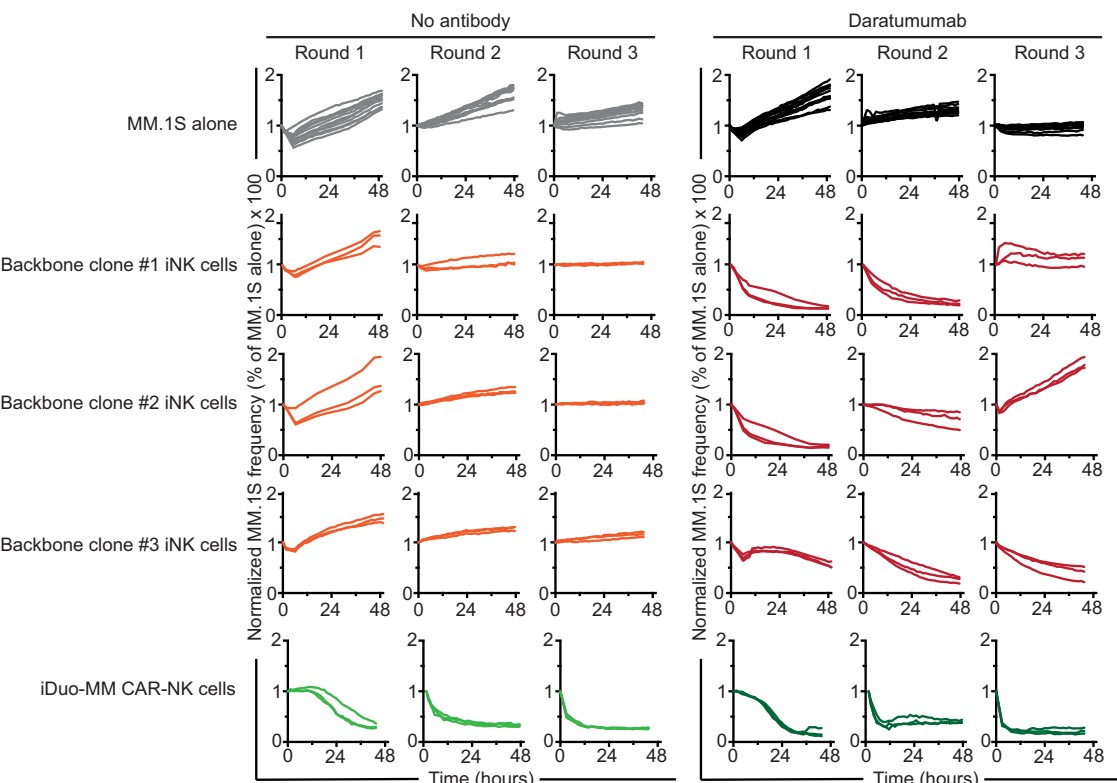

**Fig. 2 | iDuo-MM CAR-NK cells with the anti-BCMA CAR4 construct mediate sustained cytotoxic activity against MM.1S cells in a serial restimulation assay.** A serial restimulation cytotoxicity assay was performed with MM.1S cells transduced with NucLight Red as targets in the presence or absence of daratumumab, using backbone iNK cells or iDuo-MM CAR-NK cells as effectors. MM.1S cells were plated on the day of assay initiation followed by the addition of iNK cells at a 1:1 E:T ratio. After 48 hours of co-culture (round 1), non-adherent effector cells were transferred to new plates containing freshly plated MM.1S target cells (round 2) without recalibration of the E:T ratio. This procedure was repeated for a third round (round 3) of stimulation. MM.1S target cell survival was assessed in real-time using IncuCyte imaging. Data are representative of two independent experiments. Source data are provided as a Source Data file.

CAR-NK cells, comparable to PBNK cells with mean values of ~3% and 1%, respectively, at the highest E:T ratio (Fig. 3b).

In addition to cellular cytotoxicity, NK cells contribute to antiviral and antitumor responses through the release of inflammatory cytokines including tumor necrosis factor (TNF) and interferon (IFN)-γ. To determine whether activation through the anti-BCMA CAR and hnCD16 stimulated inflammatory cytokine production, expanded PBNK cells, and iDuo-MM CAR-NK cells were cultured alone, with daratumumab, with RPMI-8226 MM cells, or with both RPMI-8226 cells and daratumumab for 6 hours. Supernatants from all cultures were collected, and the levels of TNF and IFN-γ were measured using the MesoScale Diagnostics electrochemiluminescence platform. TNF production by expanded PBNK cells was low in response to RPMI-8226 cells and increased only modestly with the addition of daratumumab. In contrast, iDuo-MM CAR-NK cells produced high levels of TNF when co-cultured with MM targets, and the addition of daratumumab further enhanced TNF production. A similar pattern was observed in the analysis of IFN-γ concentrations in the culture supernatants (Fig. 3c). Thus, iDuo-MM CAR-NK cells mediate strong cytotoxic and inflammatory responses against BCMA+ MM cells without off-target activity against allogeneic PBMCs. To assess cytotoxic potential against hematopoietic progenitor cells (HPCs), we isolated CD34+CD38+ HPCs from umbilical cord blood and cultured these cells alone or with iDuo-MM CAR-NK cells at 0.5:1 E:T ratio in the presence or absence of daratumumab. No significant cytotoxicity against HPCs was observed, alone or in combination with daratumumab (Supplementary Fig. 1).

To assess the cytotoxic function of iDuo-MM CAR-NK cells against primary MM, we collected bone marrow biopsies from 2 patients with relapsed disease. CD138+ MM cells were enriched by magnetic bead selection and cultured alone or with iDuo-MM CAR-NK cells at a 2:1 E:T

ratio. We also assessed CD56 on patient CD138+ MM cells[30], as the lack of CD56 on the cell-surface is associated with significantly worse prognosis[31]. While assessment of patient 1 revealed the presence of a large population CD56[low/–] MM cells, co-culture with iDuo-MM CAR-NK cells resulted in a near total elimination of MM cells, supportive of a highly active CAR function. Co-culture of patient 2 CD138+ MM cells with iDuo-MM CAR-NK cells resulted in an approximately 4-fold decrease in MM cells, however, a residual quantity was still present. To test the value of the dual-targeting approach facilitated by iDuo-MM CAR-NK cells, we included daratumumab in the co-culture and showed that the addition of antibody further reduced the presence of MM cells to below one percent (Fig. 3d). Together, these results demonstrate the robust effector function and the value of the dual-targeting feature of iDuo-MM CAR-NK cells against MM cell lines and primary MM cells.

### iDuo-MM CAR-NK cells exhibit superior cytotoxicity in serial restimulation assays

The durability of the cytotoxic activity of iDuo-MM CAR-NK cells, both alone and in combination with daratumumab, was further evaluated in serial restimulation assays using MM.1S cells as targets. Four independent batches of iDuo-MM CAR-NK cells were compared against expanded PBNK cells from three healthy donors. Effector cells were co-cultured with MM.1S cells at 3:1 and 10:1 E:T ratios in the presence or absence of daratumumab without the addition of exogenous cytokines. While expanded PBNK cells exhibited strong ADCC during round 1, direct cytotoxicity was modest (Fig. 4a). Minimal PBNK cell cytotoxicity was observed with or without daratumumab during the second and third challenges with MM.1S cells at either E:T ratio. In contrast, iDuo-MM CAR-NK cells were highly cytotoxic against MM.1S cells at both E:T ratios in the presence and absence of daratumumab

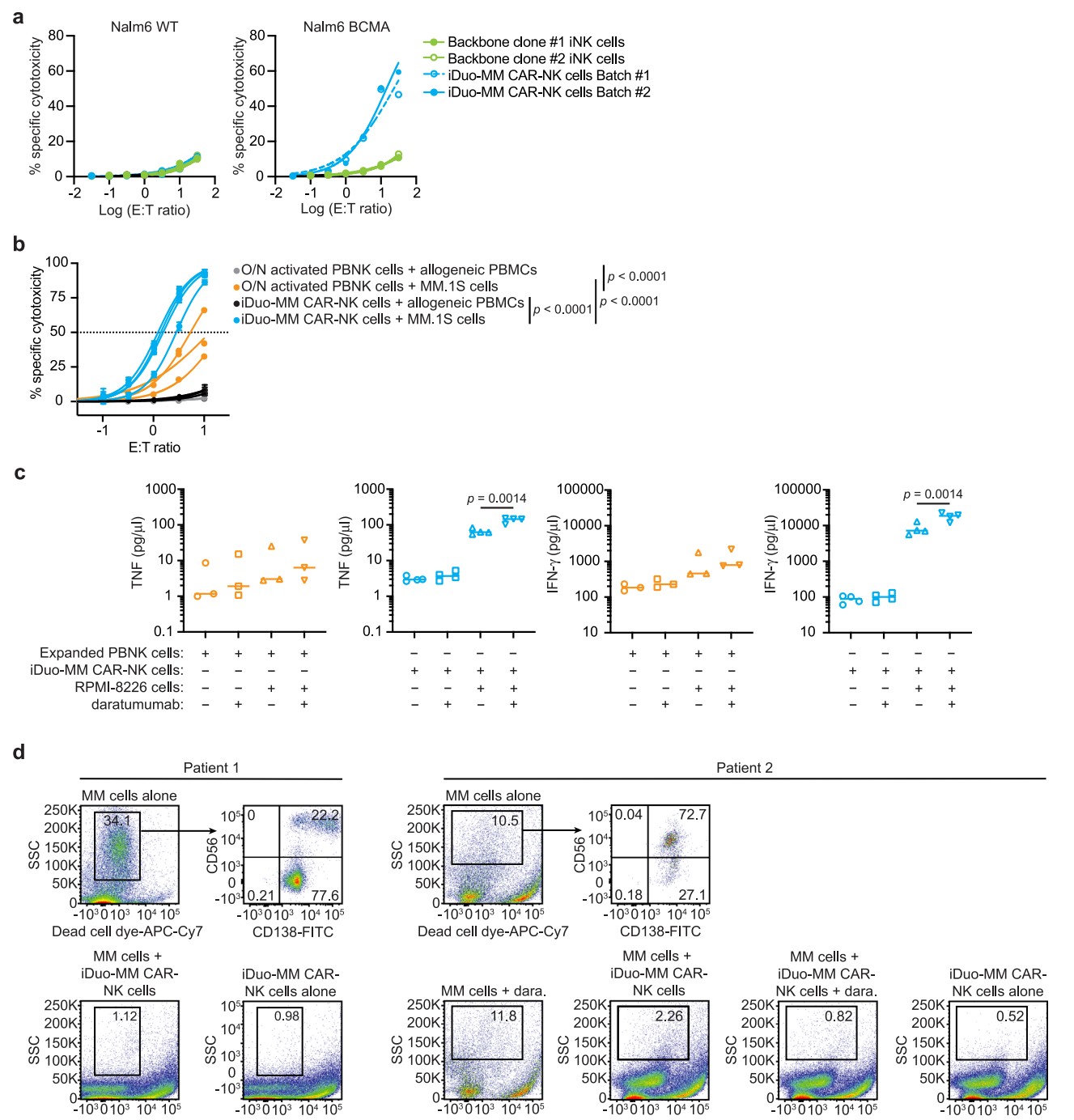

**Fig. 3 | iDuo-MM CAR-NK cells exhibit robust specific cytotoxicity and enhanced inflammatory cytokine production when combined with daratumumab. a** Two independent batches of iDuo-MM CAR-NK cells were assessed alongside iNK cells differentiated from backbone clones 1 and 2 for BCMA-specific cytotoxicity. Nalm6 cells overexpressing BCMA were mixed at a 1:1 ratio with WT Nalm6 lacking BCMA and used as targets in a 4-hour cytotoxicity assay at E:T ratios ranging from 0.03:1 to 31:1. Specific cytotoxicity was calculated independently for BCMA-negative WT Nalm6 cells (left) and BCMA-engineered Nalm6 cells (right) as the frequencies of target cells with active caspase-3/7 activity by flow cytometry. Data are from two independent experiments. **b** iDuo-MM CAR-NK cells from four independent differentiations and overnight (O/N) primed (10 ng/ml IL-15) PBNK cells isolated from three healthy donors were co-cultured with a mixture of normal donor PBMCs and MM.1S myeloma cells. Specific cytotoxicity against each cell type was assessed using a fluorescent caspase-3/7 reporter following a 4-hour incubation period. Statistical significance was determined by two-way ANOVA. Data are

presented as mean values±SD. **c** The same batches of iDuo-MM CAR-NK cells ($n = 4$) and overnight primed PBNK cells ($n = 3$) were co-cultured with RPMI-8226 myeloma cells in the presence or absence of daratumumab for 6 hours. Control conditions included effector cells cultured alone and in the presence of daratumumab only. Supernatants were collected at the end of the co-culture period, and the levels of TNF and IFN-γ were measured using the MesoScale Diagnostics (MSD) platform. Statistical significance was determined with two-sided paired Student's *t* tests. **d** Bone marrow biopsy samples were collected from two patients with relapsed MM. CD138+ MM cells were enriched by magnetic selection and co-cultured with iDuo-MM CAR-NK cells with and without daratumumab (dara) for 4 hours. No daratumumab conditions were included for functional assays with MM cells from patient 1 because of limited cell numbers. The percentages of remaining MM cells in each condition were assessed by flow cytometry. Source data are provided as a Source Data file.

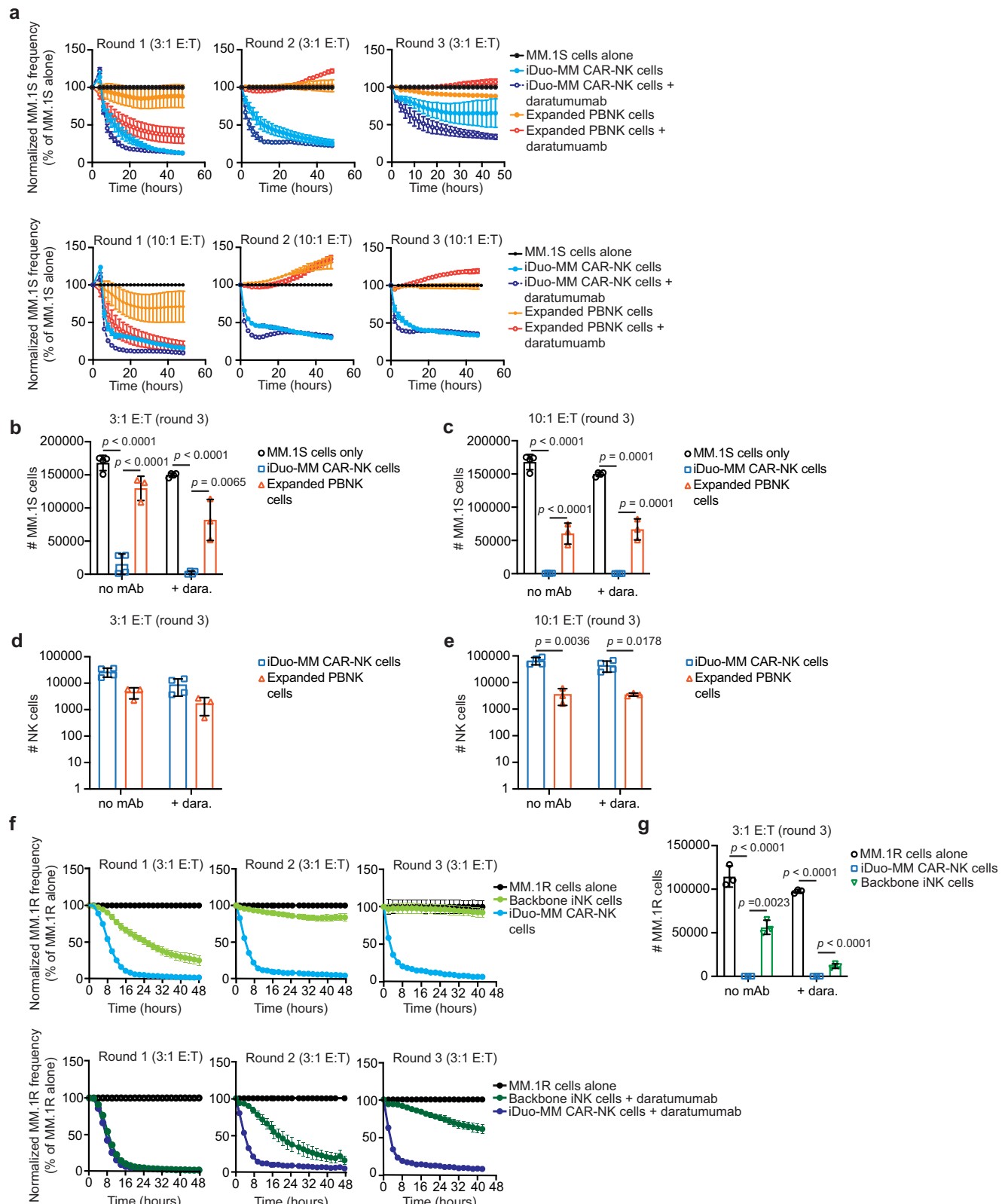

during the first and second rounds of the assay. During the third round of the restimulation assay, a distinction was observed at the lower E:T ratio between the iDuo-MM CAR-NK cell conditions, where durable tumor control required the dual-target approach supported by daratumumab. However, at the higher E:T ratio, iDuo-MM CAR-NK cells mediated robust cytotoxic responses in the absence of daratumumab,

demonstrating the importance of dual-targeting in settings where NK cell numbers are limited (Fig. 4a).

To confirm these results and determine statistical significance, the numbers of remaining MM.1S cells and effector cells were determined by flow cytometry at the conclusion of the assay. Counts of MM.1S cells from the final round of restimulation at the 3:1 E:T ratio confirmed the

**Fig. 4 | iDuo-MM CAR-NK cells display more durable cytotoxicity and persistence relative to expanded PBNK cells in serial restimulation assays. a** Serial restimulation assays were performed using NucLight Red-transduced MM.1 S cells as targets with iDuo-MM CAR-NK cells ($n = 4$ independent batches) and expanded PBNK cells ($n = 3$ healthy donors) as effectors at 3:1 and 10:1 E:T ratios in the presence and absence of daratumumab. Cytotoxicity was determined using IncuCyte imaging in three consecutive rounds of co-culture, and target cell frequencies were normalized at each time point to the target cell alone condition. At the end of the assays, cells were collected and analyzed by flow cytometry to determine the numbers of MM.1 S cells remaining in the **b** 3:1 E:T and **c** 10:1 E:T co-cultures and the numbers of NK cells remaining in the **d** 3:1 E:T and **e** 10:1 E:T co-cultures. **f** Serial restimulation assays were also performed with NucLight Red-transduced MM.1R cells as targets backbone iNK cells ($n = 3$ independent batches) and iDuo-MM CAR-NK cells ($n = 3$ independent batches) as effectors at 3:1 E:T ratios. **g** Flow cytometry was used to determine the numbers of MM.1R cells remaining in each co-culture condition. Statistical significance was determined using multiple unpaired $t$ tests using the False Discovery Rate approach. All data are presented mean values ± SD. Source data are provided as a Source Data file.

IncuCyte results, as iDuo-MM CAR-NK cells significantly reduced MM.1S cell counts compared with MM.1S cells in control wells (Fig. 4b). The addition of daratumumab led to a further reduction in the number of remaining MM.1S cells. In contrast, expanded PBNK cells were less effective at reducing MM.1S cell counts alone or with daratumumab (Fig. 4b). At the 10:1 E:T ratio, iDuo-MM CAR-NK cells alone nearly eradicated MM.1S cells in culture. Again, expanded PBNK cells were less cytotoxic against MM.1S cells, and daratumumab had no beneficial effect (Fig. 4c). At the 3:1 E:T ratio, iDuo-MM CAR-NK cells also exhibited greater persistence than expanded PBNK cells, with counts that were 5-fold higher with or without daratumumab (Fig. 4d). At the 10:1 E:T ratio, iDuo-MM CAR-NK cell numbers were 12- and 18-fold higher than expanded PBNK cells in the presence and absence of daratumumab, respectively (Fig. 4e). We also assessed surface levels of the checkpoint inhibitory receptors LAG-3 and TIM-3 on expanded PBNK cells and iDuo-MM CAR-NK cells at the end of the assay to look for any obvious differences between PBNK and iNK cells. A small fraction of PBNK cells expressed LAG-3, which was further increased with the addition of daratumumab. LAG-3 was not present at reliably detectable levels of iDuo-MM CAR-NK cells. TIM-3 was observed on larger percentages of PBNK cells and iDuo-MM CAR-NK cells and was moderately increased in the presence of daratumumab (Supplementary Fig. 2).

Restimulation experiments were also performed with MM.1R cells as targets using 3 independent batches of backbone iNK cells and iDuo-MM CAR-NK cells as effectors at 3:1 E:T ratios with or without the addition of daratumumab. iDuo-MM CAR-NK cells maintained robust cytotoxicity throughout all 3 rounds of target engagement with and without daratumumab. Backbone iNK cells exhibited strong cytotoxicity in the first round of the assay, particularly when combined with daratumumab, but progressively lost killing capacity during rounds 2 and 3 (Fig. 4f). Quantification of cells in wells after the third round of the assay showed that iDuo-MM CAR-NK cells killed nearly all MM.1R cells and that robust killing by backbone iNK cells required the addition of daratumumab (Fig. 4g). These results highlight the importance of the engineered modalities in enhancing NK cell-mediated cytotoxicity and persistence.

## iDuo-MM CAR-NK cells control MM and persist at high levels in vivo

Having thoroughly characterized the function and persistence of iDuo-MM CAR-NK cells in vitro, we next wanted to assess the in vivo efficacy of these cells. To this end, MM.1 S cells expressing firefly luciferase (MM.1S-luc) were injected i.v. into NSG mice. After allowing the tumor to establish, groups of mice received no treatment, daratumumab alone, 1 dose of iDuo-MM CAR-NK, or backbone iNK cells in the absence or presence of daratumumab. Additional groups of mice received 3 doses of backbone iNK cells, iDuo-MM CAR-NK cells, or expanded PBNK cells immediately thawed from cryopreservation (Fig. 5a). Tumor growth was monitored by bioluminescence imaging (BLI) out to day 31 post-engraftment. In the single effector dose part of the study, treatment with daratumumab or iDuo-MM CAR-NK cells alone had similarly moderate effects, with iDuo-MM CAR-NK cells initially having a more prominent antitumor effect. Backbone iNK cells were ineffective alone and demonstrated significant antitumor function when combined with daratumumab. However, the combination of iDuo-MM CAR-NK cells and daratumumab nearly eliminated all detectable MM1.S tumor cells and resulted in the most durable tumor control (Fig. 5a, b). This finding is consistent with previous studies showing moderate single-agent activity for daratumumab in immunodeficient NSG mice[7]. In addition to monitoring tumor growth, mice were bled weekly between weeks 3 and 5 for assessment of NK cell counts by flow cytometry. After single-dose administration, iDuo-MM CAR-NK cells with or without daratumumab were detected at appreciable levels in the peripheral blood at day 22 and declined to low levels by day 35 (Fig. 5c, d).

In the multi-dose part of the study, treatment with backbone iNK cells or expanded PBNK cells failed to control MM.1S tumor growth, while iDuo-MM CAR-NK cells mediated durable antitumor activity (Fig. 5e). Remarkably, both backbone iNK cells and iDuo-MM CAR-NK cells persisted at high levels in the peripheral blood of mice through day 35. The persistence of expanded PBNK cells was significantly lower, failing to reach counts >1 cell/μl at day 35 despite multiple doses (Fig. 5f, g). The high level of persistence of backbone iNK cells and iDuo-MM CAR-NK cells are attributable to the IL-15RF, and *CD38* KO signaling cascades[28,29]. However, the lack of antigen targeting through the anti-BCMA CAR on backbone iNK cells clearly impacted the ability of these cells to control tumor burden in vivo. In additional multi-dose experiments, treatment with iNK cells harboring the *CD38* knockout edit alone failed to control MM.1S tumor growth in the absence or presence of daratumumab. The numbers of these cells were low after adoptive transfer and comparable to that of PBNK cells in this model, suggesting that both iNK cells and PBNK cells fail to persist in the absence of cytokine support (Supplementary Fig. 3). The full gating strategy used to identify human NK cells in mouse peripheral blood is shown in Supplementary Fig. 4.

## iDuo-MM CAR-NK cells maintain in vivo antitumor function without exogenous cytokines

Cytokine support is typically necessary to drive the antitumor activity of adoptively transferred allogeneic NK cells, and preclinical studies often test NK cells along with recombinant human IL-2[22]. However, systemic clinical administration of cytokines such as IL-2 and IL-15 are challenging, as they typically have a short half-life. Additionally, in vivo administration of IL-2 stimulates the activation and expansion of immunosuppressive T regulatory (Treg) cells that dampen NK cell function, and systemic administration of IL-15 promotes patient CD8+ T cell-mediated allo-rejection that can hasten donor NK cell elimination[32–34]. To compare iDuo-MM CAR-NK cell antitumor activity in the presence and absence of exogenous cytokine support, we engrafted mice with MM.1S-luc cells. After allowing the tumor to establish, groups of mice received three doses of $1 \times 10^7$ iDuo-MM NK cells from two independent research batches thawed from cryopreservation or one dose of $2 \times 10^6$ primary anti-BCMA-CAR-T cells. We limited dose administration of primary CAR-T cells to $2 \times 10^6$ cells to allow for the study to be maintained for at least 30 days since the administration of primary CAR-T cells typically results in graft versus host disease (GvHD) 4-6 weeks after adoptive transfer. Half of the mice in these treatment groups received twice-weekly injections of IL-2 and IL-15, and the other half did not receive any exogenous cytokine support. Tumor growth was assessed weekly by BLI (Fig. 6a). Treatment

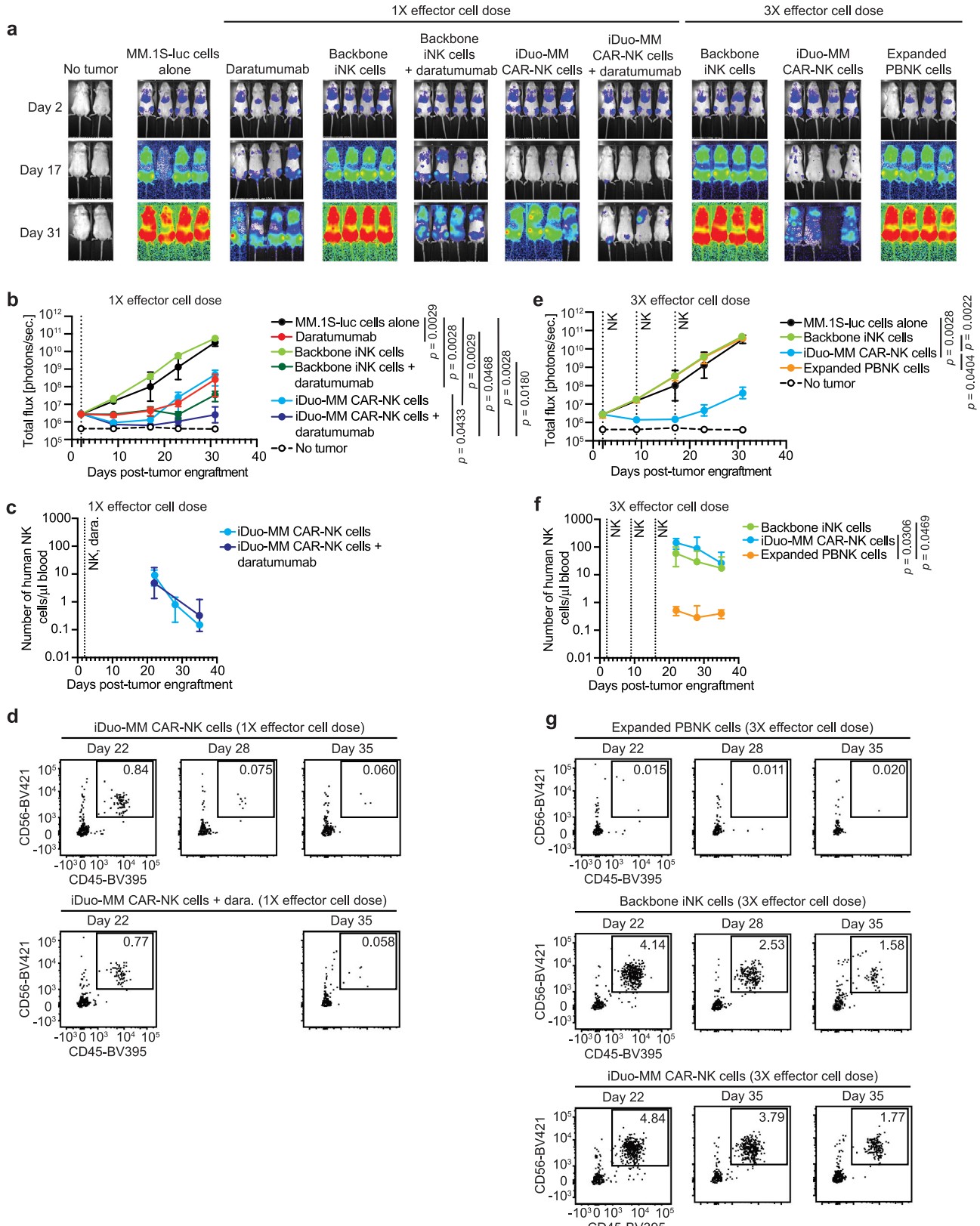

with iDuo-MM CAR-NK cells or primary anti-BCMA CAR-T cells combined with cytokine support resulted in significant tumor regression and delayed tumor outgrowth for both groups (Fig. 6a–c). Interestingly, in the study groups where cytokine support was not introduced, iDuo-MM CAR-NK cells mediated antitumor activity comparable to the counterpart treatment group that received cytokine support.

However, primary anti-BCMA-CAR-T cells demonstrated less potency in the absence of cytokines, suggesting that at the dose of $2 \times 10^6$ cells cytokine support was still required (Fig. 6a–c). Together, these results demonstrate the ability of the membrane-bound IL-15RF molecule to support durable iDuo-MM CAR-NK cell antitumor function in vivo in the absence of exogenous cytokines (Fig. 6d).

**Fig. 5 | iDuo-MM CAR-NK cells mediate superior in vivo tumor control and persist at high levels in the peripheral blood.** NSG mice ($n = 28$) were engrafted with $2 \times 10^5$ luciferase-transduced MM.1 S cells. After 2 days, groups of mice ($n = 4$/group) received no treatment, daratumumab alone, 1 i.v. injection of $1 \times 10^7$ backbone iNK cells or iDuo-MM CAR-NK cells, 1 injection of $1 \times 10^7$ backbone iNK cells or iDuo-MM CAR-NK cells with daratumumab, three injections of $1 \times 10^7$ backbone iNK cells, iDuo-MM CAR-NK cells, and expanded PBNK cells thawed from cryopreservation. Second and third effector cell injections were given at days 9 and 16 post-tumor engraftment. **a** Bioluminescence imaging of mice at days 2, 17, and 31. **b** Graphical representation of the bioluminescence data for the single effector dose

arm of the study. **c** Peripheral blood was drawn from mice at days 22, 28, and 35 for flow cytometry analysis to determine human NK cell counts for the single effector dose arm of the study. **d** Representative flow cytometry plots for analysis of mouse peripheral blood in the single effector dose experiments. **e** Graphical representation of the bioluminescence data for the multiple effector dose arm of the study. **f** Analysis of human NK cell counts in the peripheral blood for the multiple effector dose arm of the study. **g** Representative flow cytometry plots for analysis of mouse peripheral blood in the multiple effector dose experiments. Statistical significance was determined by two-way ANOVA. All graphed data are presented mean values ± SD. Source data are provided as a Source Data file.

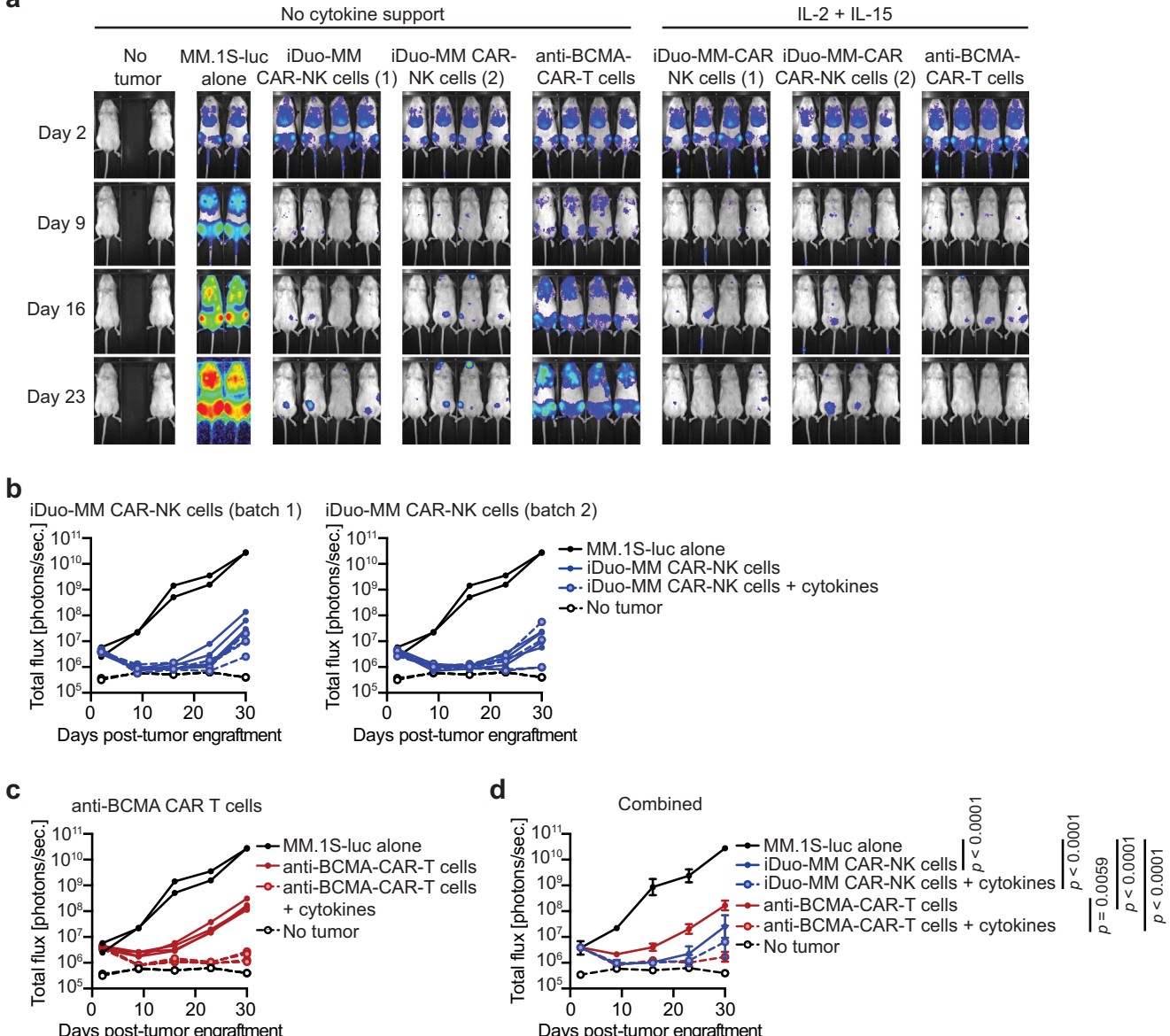

**Fig. 6 | iDuo-MM CAR-NK cells maintain in vivo antitumor function in the absence of exogenous cytokine support.** NSG mice ($n = 26$) were engrafted with $2 \times 10^5$ luciferase-transduced MM.1S cells. Groups of mice ($n = 4$/group) received either no treatment or i.v. injections of $1 \times 10^7$ iDuo-MM CAR-NK cells thawed from cryopreservation on days 2, 9, and 16 post-tumor engraftment. Additional groups of mice were given i.v. injections of $2 \times 10^6$ primary anti-BMCA-CAR-T cells on day 2. Two independent batches of iDuo-MM CAR-NK cells were used, and all three treated groups were split and either given no exogenous cytokine support or supplemented with twice-weekly injections of IL-2 and IL-15. **a** Bioluminescence

imaging of mice at days 2, 9, 16, and 23 post-tumor engraftments. **b** Graphical representation of bioluminescence data comparing mice treated with both batches of iDuo-MM CAR-NK cells to mice with tumor alone. **c** Graphical representation of bioluminescence data comparing mice treated with anti-BCMA-CAR-T cells to mice with tumor alone. **d** Graphical representation of bioluminescence data from all treatment and control groups. Statistical significance was determined by two-way ANOVA. Data are presented mean values ± SD. Source data are provided as a Source Data file.

### iDuo-MM CAR-NK cells can be combined with daratumumab and γ-secretase inhibition for enhanced tumor control

To assess dual-antigen responses in vivo more thoroughly, we tested the combination of iDuo-MM CAR-NK cells and daratumumab. This approach takes advantage of the enhanced ADCC driven by the hnCD16 transgene and the prevention of anti-CD38 mAb-mediated fratricide by *CD38* knockout. Additionally, BCMA can be directly cleaved by γ-secretase, and inhibition of γ-secretase in vivo enhances BCMA surface density[21]. While our BCMA binder can elicit a CAR-mediated response at low antigen densities[27], we wanted to determine whether increasing BCMA surface density on MM cells through prevention of γ-secretase-mediated cleavage could impact iDuo-MM CAR-NK cell activity. To validate the efficacy of the γ-secretase inhibitor (GSI) LY3039478 in elevating BCMA surface density on MM.1S cells in vivo, LY3039478 was dosed orally in NSG mice engrafted with MM.1S-luc cells. Mice were sacrificed 8, 24, and 48 hours post-GSI dosing, and bone marrow was harvested for assessment of BCMA surface levels on MM.1S-luc cells by flow cytometry. We observed a potent, but transient increase in BCMA levels on MM.1S cells within the first 24 hours after GSI administration (Fig. 7a, b). Next, we engrafted NSG mice with MM.1S-luc cells and allowed tumors to establish. To increase the resilience of the model, groups of mice received three lower doses of $5 \times 10^6$ iDuo-MM CAR-NK cells thawed from cryopreservation and directly administered or a single dose of $2 \times 10^6$ primary anti-BCMA-CAR-T cells. Selected groups of mice received 3 doses of daratumumab either alone or in combination with iDuo-MM CAR-NK cells. Additional groups of mice received GSI orally twice per week for 3 weeks. Tumor growth was monitored by BLI (Fig. 7c). GSI treatment did not impact tumor growth, while daratumumab reduced tumor burdens early during the experiment. Both iDuo-MM CAR-NK cells and primary anti-BCMA-CAR-T cells exhibited tumor control that was further improved by GSI treatment. The addition of daratumumab significantly enhanced iDuo-MM CAR-NK cell antitumor activity, and the combination of iDuo-MM CAR-NK cells, GSI, and daratumumab resulted in the deepest and most sustained anti-MM responses (Fig. 7d, e). Peripheral blood NK and T cell counts were tracked between days 6 and 18 in this study. No differences in counts between iDuo-MM CAR-NK cells and anti-BMCA CAR-T cells were observed (Fig. 7f). Together, these results demonstrate the in vivo dual functionality of iDuo-MM CAR-NK cells treated in a manner resembling in an off-the-shelf administration and the benefit of antigen stabilization for a superior antitumor response.

## Discussion

The therapeutic landscape for the treatment of MM has changed dramatically during the past decade with the introduction of several new agents[3–11]. Despite these effective treatments, relapse is inevitable, requiring sustained therapy to control disease progression[35]. Primed immune responses are often insufficient to control MM due to the suppressive tumor microenvironment[36]. Nonetheless, immune-mediated eradication of MM holds considerable promise. A subset of myeloma patients undergoing allogeneic hematopoietic cell transplantation (HCT) achieved durable remissions, demonstrating the potential of immunotherapy in this disease setting[37]. Unfortunately, this success was accompanied by high treatment mortality rates and inconsistent benefit observed across randomized trials[38–40]. In the past several years safer and more effective approaches, such as CAR-T cells, have been developed to treat myeloma patients[12–17]. Despite the promising efficacy of CAR-T cells, more than 50% of patients relapse in less than 1 year, and severe adverse events were seen in some patients[15]. There are also major limitations related to CAR-T cell access including cost and complex manufacturing resulting in inconsistent and variable final products. There is often a time lag between cell collection, manufacturing, and delivery of treatment that can result in disease progression and death before a patient could receive treatment. Thus, off-the-shelf approaches that are effective, safe, lower in cost, and provided on demand are needed for cellular immunotherapy to reach its full potential[41].

Compared to T cells, NK cells are promising candidates for allogeneic off-the-shelf immunotherapy due to their preferable safety profile with low risks for graft versus host disease (GvHD), CRS, and neurotoxicity[22,23]. However, there are also limitations of peripheral blood NK cell therapy. Historically, only a single-cell dose could be collected from a single related donor for use in adoptive transfer. The export of NK cells collected from a donor apheresis is difficult and inconsistent. Furthermore, genetic editing of NK cells is very difficult, which has significantly impeded its therapeutic potential. These challenges have limited the ability to successfully test NK cell immunotherapy in the cancer setting. We have developed a culture system for generating large numbers of iNK cells[24,28]. The manufacturing process starts with a clonal population of multiplexed engineered iPSCs that can be accessed as a renewable source of starting material to generate large-scale off-the-shelf therapeutic doses of iNK cells that are stored and made available on demand. These iNK cells are transcriptionally similar to peripheral blood NK cells, maintain stability as frozen banks, and exhibit broad cytotoxic function through both innate and engineered receptors upon thaw without the need for recovery or cytokine priming, which has been a requirement for most NK cell products. Additionally, iNK cells activate T cells and make them more responsive to programmed death 1 (PD-1) blockade to induce the second wave of immune response for enhanced tumor elimination[24].

One advantage of the iNK cell platform is that precise multi-gene engineering can be efficiently performed at the iPSC stage to avoid the complications associated with the genetic editing of NK cells. Using an in vivo xenograft model of human B cell lymphoma, we previously demonstrated that iNK cells engineered with hnCD16 in combination with an anti-CD20 monoclonal antibody led to significantly improved tumor regression compared with anti-CD20 antibody combined with PBNK cells or non-transduced iNK cells[42]. We have also shown that *CD38* knockout, in addition to preventing daratumumab-mediated fratricide, alters iNK cell metabolism and protects against oxidative stress[28].

Here, we describe quadruple gene-modified iDuo-MM CAR-NK cells containing an anti-BCMA CAR, hnCD16, IL-15RF, and *CD38* knockout (Fig. 1A). iDuo-MM CAR-NK cells were designed for maximal targeting of MM cells through the anti-BCMA CAR and via the engagement of hnCD16 with anti-CD38 mAbs, including daratumumab, for a unique dual-targeting approach. This effort began with phenotypic and functional characterizations of multiplexed engineered NK cells derived from iPSC clones genetically edited with hnCD16 and IL-15RF transgenes and knockout of *CD38*. These cells were further engineered with an anti-BCMA CAR construct. The top performing clone was selected and termed iDuo-MM CAR-NK cells. Unlike PBNK cells, iDuo-MM CAR-NK cells exhibited sustained and durable tumor control in both in vitro restimulation assays and in vivo xenogeneic adoptive transfer experiments. Importantly, while iDuo-MM CAR-NK cells demonstrated antigen-specific antitumor activity against MM cells, they maintained their natural ability to avoid off-target cytotoxicity towards healthy cells. The antitumor efficacy of iDuo-MM CAR-NK cells was comparable to that of primary anti-BCMA CAR-T cells. However, complications such as GvHD were not observed for iDuo-MM CAR-NK cells. In co-culture assays with primary MM cells from 2 patients, iDuo-MM CAR-NK cells effectively eliminated MM cells from one of the patients. Residual MM cells were present from the other patient after co-culture with iDuo-MM CAR-NK cells that were eliminated with the addition of daratumumab. Heterogeneity in surface BCMA levels on MM cells from different individuals could impact the efficacy of iDuo-MM CAR-NK cells as a monotherapy and reinforces the rational for combination therapy with daratumumab.

In addition to potent CAR-mediated signaling, it is clear from our data that membrane-bound IL-15RF promotes iDuo-MM CAR-NK cell

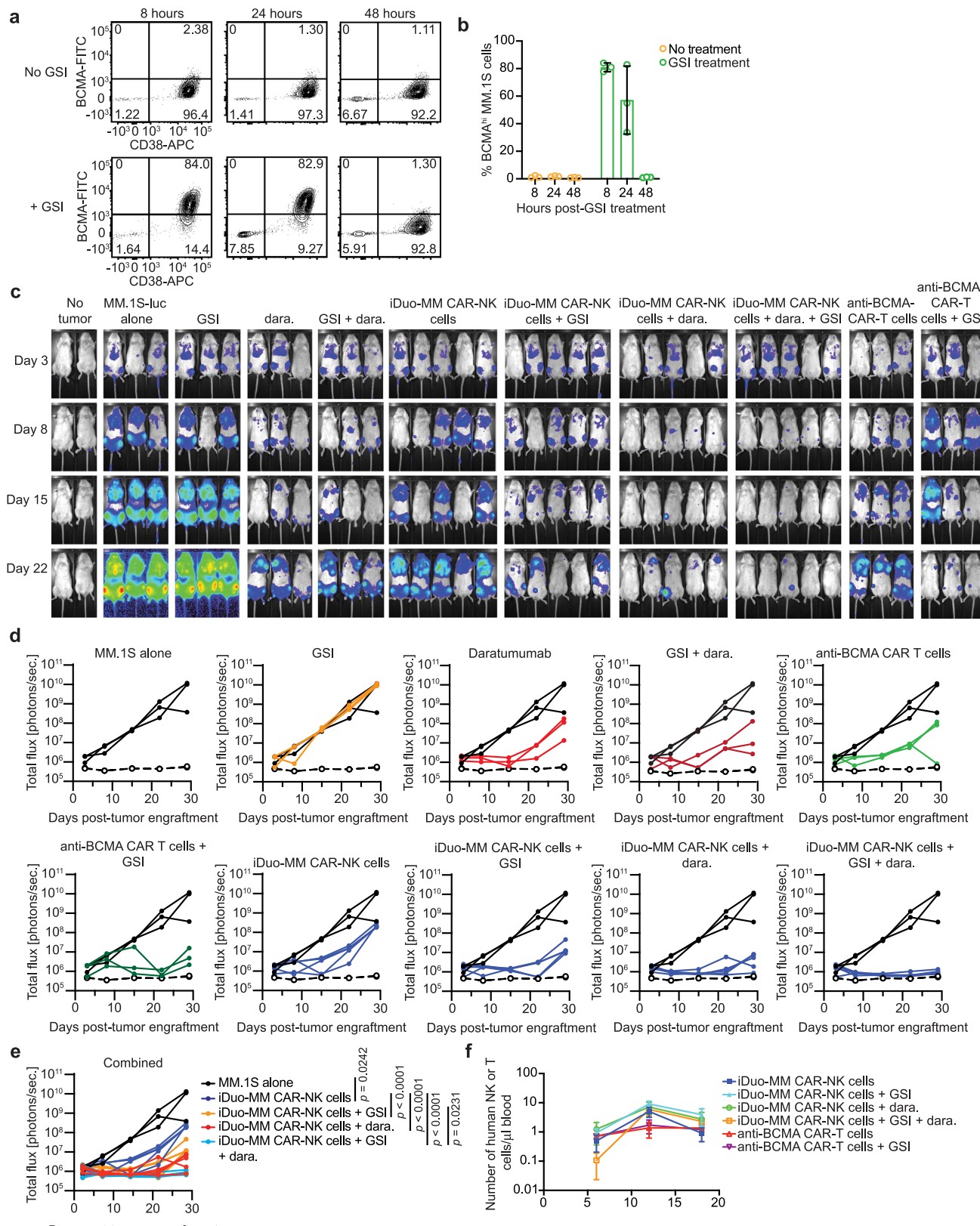

persistence and function without the need for additional cytokine support in the multi-dose setting. This was evident in xenogeneic adoptive transfer experiments comparing adoptive transfer of iDuo-MM CAR-NK cells with or without exogenous cytokine support. One caveat to consider is that, despite expression of IL-15RF, iDuo-MM CAR-NK cells displayed moderate persistence in the peripheral blood of

mice after a single injection. This may be due to migration of the cells out of the blood and into other tissues. Multiple injections of cells appear to be necessary to sustain adoptively transferred iNK cells in circulating peripheral blood. The reason for differences in persistence in blood based on dose number and migration to various tissues is currently under investigation. In contrast, PBNK cells and iNK cells

**Fig. 7 | Dual targeting in combination with daratumumab and BCMA stabilization via γ-secretase inhibition enhance iDuo-MM CAR-NK cell in vivo antitumor activity.** NSG mice engrafted with $2 \times 10^5$ luciferase-transduced MM.1S cells ($n = 18$) were orally dosed with LY3039478 (1 mg/kg) or left untreated. Groups of mice ($n = 3$/group) were sacrificed after 8, 24, and 48 hours. Bone marrow was collected for analysis of BCMA surface density on MM.1S cells. **a** Representative flow cytometry plots showing BCMA levels on MM.1S cells from mice with and without GSI treatment. **b** Graphical representation of the cumulative data. Data are presented mean values ± SD. **c** NSG mice ($n = 38$) were engrafted with $2 \times 10^5$ luciferase-transduced MM.1S cells. Groups of mice ($n = 3$–5/group) received either no treatment or i.v. injections of $5 \times 10^6$ iDuo-MM CAR-NK cells thawed from cryopreservation on days 3, 10, and 17. Separate groups of mice received i.v.

injections of $2 \times 10^6$ primary anti-BCMA CAR-T cells on day 3. Some groups of mice were orally dosed with GSI twice weekly for 3 weeks beginning 1 day before MM.1S engraftment. Some groups of mice were administered daratumumab (dara) along with each iDuo-MM CAR-NK cell injection. Shown is bioluminescence imaging of mice at days 3, 8, 15, and 22. **d** Graphical representation of bioluminescence data comparing each treatment group to the MM.1S tumor alone group. **e** Graphical representation of all treatment groups along with the MM.1S tumor alone group. **f** Mouse peripheral blood was collected on days 6, 12, and 18 for assessment of human NK and T-cell numbers by flow cytometry. Shown are cumulative data with three mice analyzed per treatment group at each time point. Statistical significance was determined by two-way ANOVA. Data are presented mean values ± SD. Source data are provided as a Source Data file.

lacking IL-15RF required exogenous cytokines, and their removal dramatically reduced their antitumor efficacy and the durability of their responses to MM cells. Interestingly, the performance of primary CAR-T cells also improved with the addition of cytokine support. These results have important implications for the use of NK cells in immunotherapy where the goal is to avoid supplemental dosing of IL-2 that expands suppressive Tregs[32] or IL-15 that activates alloreactive CD8+ T cells[33,34]. While iDuo-MM CAR-NK cells alone were effective at controlling MM tumor growth in vivo, deeper, and more sustained antitumor efficacy was achieved with a dual-targeting approach.

When combined with daratumumab, iDuo-MM CAR-NK cells virtually eliminated MM cells through ADCC mediated by hnCD16 in both in vitro restimulation assays and an in vivo xenograft model. Unique to this maximized ADCC activity is the knockout of *CD38* to avoid anti-CD38 mAb-mediated fratricide. Notably, the dual-antigen cytotoxicity feature of iDuo-MM CAR-NK cells was further enhanced with a BCMA antigen stabilization approach using γ-secretase inhibition, resulting in complete elimination of the tumor burden in the MM xenograft model. Taken together, the preclinical data presented here combined with the off-the-shelf availability, cost-effectiveness, and uniformity of product features support the translation of iDuo-MM CAR-NK cells (FT576) in clinical trials to treat patients with refractory and relapsed MM (NCT05182073). FT576 as a monotherapy or in combination with daratumumab has begun in the dose escalation trial starting at a single dose of 100 million cells.

## Methods
### Study approvals
NOD.Cg-*Prkdc*scid*IL2rg*tm1Wjl/SzJ (NSG) mice (Jackson Laboratories) were housed in the institution's Association for Assessment and Accreditation of Laboratory Animal Care (AAALAC)-accredited animal care facility at Fate Therapeutics Torry Pines campus. All experiments were reviewed and approved by Fate Therapeutics Animal Care Committee (IACUC) under the protocol 2019-11-01 O'Rouke. Patient MM cells were obtained from bone marrow biopsies performed at the Clinics and Surgery Center on the East Bank campus of the University of Minnesota with approval from the institutional review board (IRB).

### Study design
The objective of this study was to engineer an iNK cell product with robust CAR-mediated direct cytotoxicity and indirect ADCC responses against MM that maintained antitumor function and persisted at high levels in vivo. Toward this objective, we used our established methods for generating high-quality iPSCs, performing multi-gene engineering, and differentiating these cells into CD34+ hematopoietic progenitor cells[28,40–43]. Hematopoietic progenitors were then differentiated along the NK cell lineage using stroma cells and media preparations[24,44]. This system has been scaled up to generate clinical-grad cells in the GMP facilities at the University of Minnesota and Fate Therapeutics. The antitumor efficacy of iDuo-MM NK cells was studied extensively with flow cytometry-based specific cytotoxicity assays and live imaging

of cytotoxicity using IncuCyte Live Cell Analysis Systems (Essen BioScience). Xenograft adoptive transfer experiments were performed to assess the ability of iDuo-MM NK cells to exert antitumor function and persist in the peripheral blood independent of exogenous cytokines.

### iNK cell culture and phenotyping
iPSCs were differentiated towards the mesoderm and CD34+ hematopoietic progenitor stages in StemPro34 media (Thermo Fisher) supplemented with BMP4 and bFGF (Life Technologies). iCD34+ cells were subsequently enriched prior to differentiation. At the beginning of the iNK cell differentiation culture, iCD34+ cells were plated on OP9 cells in B0 media[43] to support NK cell differentiation from hematopoietic progenitors. After 20-30 days of culture, iNK cells were harvested, co-cultured with irradiated K562 cells transduced with membrane-bound IL-21 and 4-1BB ligand (4-1BBL) constructs in supplemented B0 media for 2 weeks. PBNK were expanded using the same feeder cells for 2 weeks. Fold expansion rates of iDuo-MM CAR-NK cells from the iPSC stage through iNK cell differentiation and expansion are shown in Supplementary Fig. 5.

iDuo-MM CAR iPSCs were created by CRISPR-mediated target integration followed by lentiviral transduction. The donor plasmid contained IL-15RF and hnCD16 in a 2A peptide-connected bi-cistronic expression cassette flanked by 2 homology arms to facilitate targeted integration within the *CD38* locus. IL-15RF was constructed by combining IL-15 (GenBank accession # NM_000585) and IL-15 receptor alpha (GenBank accession # NM_002189) and hnCD16 was constructed as described previously[26]. For targeted integration at *CD38*, AsCpf1 nuclease (Aldevrong) and a targeting gRNA (5'-TCCCCGGACACC GGGCTGAAC-3'), were mixed and incubated for 10 minutes to form ribonucleoprotein, followed by an addition of the donor plasmid and iPSCs. Transfection was performed using a Neon Transfection System 100 µl Kit (Thermo Fisher) following the manufacturer's protocol. A single-cell-derived iPSC clone was isolated, confirmed for accurate gene targeting, and transduced with VSV-G pseudotyped lentiviruses containing the anti-BCMA-CAR4. The anti-BCMA single chain variable fragment (scFv) and CAR4 design have been reported[27,29]. All iDuo-MM CAR-NK cells were from a single-cell origin with clonal CAR vector integration. Fluorescently conjugated antibodies specific for the following human epitopes were used for flow cytometry: CD3 (OKT3), CD56 (HCD56), DNAM-1 (11A8), NKp44 (P448), NKG2D (1D11), CD16 (3G8), CD38 (HIT2), IL-15R (JM7A4), CD138 (MI15), BCMA (19F2), LAG-3 (7H2C65), TIM-3 (F38-2E2), CD34 (581), CD45 (HI30) (all BioLegend), NKp30 (P30-15; BD Biosciences), KIR2DL1 (FFAB1844P; R&D Systems), and KIR2DL2/3 (FAB1848P; R&D Systems). To discriminate live populations, cells were stained with the LIVE/DEAD Fixable Near-IR Dead Cell Stain (Thermo Fisher). For detection of the anti-BCMA CAR, cells were stained with BCMA-biotin (BCA-H82E4; AcroBiosystems) followed by PE-streptavidin (554061; BD Biosciences). Flow cytometry was performed on Fortessa and LSRII instruments (BD Biosciences). Flow cytometry data were analyzed with FlowJo software (v10.7.1) (BD Biosciences).

## Isolation and expansion of peripheral blood NK cells

Mononuclear cells were isolated from blood products by density gradient centrifugation using Ficoll-HiPaq (GE Healthcare). NK cells were then enriched using EasySep Human NK Cell Enrichment Kits (StemCell Technologies) per the manufacturer's instructions. For overnight priming, NK cells were incubated in media containing 10 ng/ml IL-15 (PeproTech). For ex vivo expansion, peripheral blood NK cells were co-cultured with irradiated, gene-modified K562 cells for 14 days in B0 media with 250 U/ml IL-2.

## Cell lines

MM.1S (CRL-2974), MM.1R (CRL-2975), RPMI-8226 (CCL-155), Nalm6 (CRL-3273), OP9 (CRL-2749), and K562 (CRL-3343) cells were obtained from the American Tissue Culture Collection (ATCC). All cell lines were cultured in RPMI 1640 media (Corning) supplemented with 10% fetal bovine serum (HyClone) and penicillin/streptomycin. Cells were kept at the low passage and maintained at 37 °C and 5% $CO_2$. All cell lines were tested for mycoplasma by PCR monthly. Nalm6 cells were transduced with lentivirus containing a BCMA overexpression construct. $BCMA^+$ Nalm6 cells were sorted to high purity by flow cytometry for use in function assays.

## Isolation of primary T cells and generation of anti-BCMA-CAR-T cells

For the collection of T cells, PBMCs were first isolated by density gradient centrifugation using Ficoll-HiPaq. T cells were then enriched using the EasySep Human T Cell Isolation Kit (StemCell Technologies) per the manufacturer's instructions. For the generation of anti-BCMA-CAR-T cells, enriched $CD3^+$ T cells were stimulated with anti-CD3/CD28 dynabeads (Gibco) for 25 hours. Activated T cells were transduced with lentivirus supplemented with 4 µg/ml polybrene and centrifuged at $1035 \times g$ for 90 minutes at 32 °C. Beads were removed by magnetic selection on day 4. Transduced T cells were continuously expanded in media containing 50 U/ml IL-2 for an additional 4 days before harvesting for use in functional experiments.

## In vitro function NK cell function assays

For serial restimulation experiments, MM.1S transduced with NucLight Red (Sartorious) were seeded into 96-well plates (Corning). The next day, iNK cells were added at the indicated E:T ratios. Daratumumab (Janssen Pharmaceuticals) was added to select wells at a concentration of 10 µg/ml. After 48 hours of co-culture, non-adherent effector cells were transferred to new plates containing freshly plated MM.1S target cells with no recalibration of the E:T ratio and co-cultured for another 48 hours. This procedure was repeated for a third round of stimulation, and target cell survival was assessed in real-time using an IncuCyte Live Cell Analysis System (Essen BioScience). Live cell numbers were quantified with IncuCyte S3 software (Essen BioScience) and normalized to the number of live cells remaining in the target cell-only control group. For specific cytotoxicity assays, wild-type Nalm6 cells and Nalm6 cells with transgenic expression of BCMA were labeled with CellEvent Caspase-3/7 Green Detection Reagent (Thermo Fisher) as per the manufacturer's instructions. This assay was shown to yield comparable results when compared to $^{51}CR$-release assays, but with higher sensitivity[44]. Labeled cells were co-cultured with iNK cells for 4 hours at a range of E:T ratios prior to analysis by flow cytometry. Specific cytotoxicity was determined by the frequency of caspase-3/7 activity and was calculated according to the following formula: % specific cytotoxicity = 100 × (% specific death−% spontaneous death) (100−% spontaneous death), where spontaneous death is the frequency of target cells with caspase-3/7 activity alone in culture, and specific death is the frequency of target cells with caspase-3/7 activity in co-culture with effector cells. Non-linear regression was performed on log-transformed data using GraphPad Prism v10.7.1 using the log(agonist) vs. normalized response - Variable slope parameters. Similar assays were performed using MM.1S cells and allogeneic PBMCs

as targets. To assess inflammatory cytokine production, iDuo-MM CAR-NK cells and expanded PBNK cells were co-cultured with RPMI-8226 MM cells with and without daratumumab (10 µg/ml) for 6 hours. Supernatants were then collected, and the levels of TNF and IFN-γ were assessed using the MesoScale Diagnostics electrochemiluminescence platform (Meso Scale Diagnostics).

## In vivo xenogeneic adoptive transfer experiments

Test article cells were thawed from cryopreserved stock by immersion in a 37 °C water bath followed by a single wash in NK MACS media supplemented with NK MACS supplement (Miltenyi Biotech) and resuspended in complete NK MACS media. For experiments comparing in vivo tumor control mediated by expanded PBNK cells, backbone iNK cells, and iDuo-MM CAR-NK cells, 6−8-week-old NSG ($n = 36$) mice were injected i.v. with $5 \times 10^5$ luciferase-expressing MM.1S cells. Equal numbers of male and female mice were balanced between groups. After allowing the tumor to engraft for 2 days, bioluminescence imaging was performed to quantify tumor burden and balance mice evenly into groups. Groups of mice received no treatment or $1 \times 10^7$ iDuo-MM CAR-NK cells, backbone iNK cells, or expanded PBNK cells immediately thawed from cryopreservation. Mice received additional effector cell injections at days 9 and 16 post-tumor engraftments. In these experiments, mice were also bled at days 22, 28, and 35 for the determination of human NK cell counts by flow cytometry using fluorescently labeled antibodies against CD56 and CD45.

For experiments testing the in vivo antitumor efficacy of iDuo-MM CAR-NK cells in the presence and absence of cytokine support, six-to-eight-week-old NSG mice ($n = 26$) were injected i.v. with $2 \times 10^5$ luciferase-expressing MM.1S cells. Equal numbers of male and female mice were balanced between groups. After allowing the tumor to engraft for 2 days, bioluminescence imaging was performed to quantify tumor burden and balance mice evenly into 7 groups. Groups of mice received no treatment or $1 \times 10^7$ iDuo-MM CAR-NK cells from two independent differentiations or $2 \times 10^6$ primary anti-BCMA-CAR-T cells. Mice received additional iDuo-MM CAR-NK cell injections on days 9 and 16. Half of the mice in the treatment groups received twice-weekly injections of IL-2 ($2 \times 10^3$ U) and IL-15 (5 µg) for 3 weeks.

For experiments testing iDuo-MM CAR-NK cells in combination with daratumumab and γ-secretase inhibition, six-to-eight-week-old NSG mice ($n = 38$) were dosed orally with LY3039478 (MedChemExpress; 1 mg/kg) twice per week for 3 weeks starting 1 day before tumor engraftment. Equal numbers of male and female mice were balanced between groups. At day 0, mice were injected i.v. with $2 \times 10^5$ luciferase-expressing MM.1S cells. After allowing the tumor to engraft for 3 days, bioluminescence imaging was performed to quantify tumor burden and balance mice evenly into 9 groups. Groups of mice received no treatment, γ-secretase inhibitor alone, daratumumab alone, $5 \times 10^6$ iDuo-MM CAR-NK cells, or $2 \times 10^6$ primary anti-BCMA-CAR-T cells. Mice received additional injections of iDuo-MM CAR-NK cells on days 10 and 17. Some groups of mice receiving iDuo-MM CAR-NK cells also received daratumumab (8 mg/kg) along with each cell injection. In all experiments, bioluminescence imaging was performed at regular intervals to monitor tumor progression. Imaging was conducted using an IVIS Spectrum, and all images were analyzed using Living Image Software (PerkinElmer). In our IACUC protocol maximal tumor burden is reached if tumors become larger than 2.0 centimeters or if animals appear to lose weight, appear hunched, or demonstrate inactivity, poor appetite, scruffy hair. If any of these criteria are reached, animals are euthanized. These limits were not exceeded in this study.

## Statistics and reproducibility

Statistical analyses were performed using GraphPad Prism 9 software (GraphPad). Statistical significance for bioluminescence and NK cell count data for all xenogeneic adoptive transfer experiments was determined using two-way ANOVA. Statistical significance for serial

restimulation assays was determined using multiple unpaired *t* tests using the False Discovery Rate approach. Two-sided paired Student's *t* tests were used to determine statistical significance in experiments assessing cytokine levels. Studies were planned with the minimum numbers of animals per treatment group to reproducibly observe statistically significant differences ($n = 4$–5 mice per arm per experiment). All murine experiments were replicated at least twice using iNK cells, PBNK cells, or primary CAR-T cells from different cultures. Tumor engraftment was defined by baseline BLI before effector cell adoptive transfer, and mice were randomized into treatment groups. No data were excluded at any point after adoptive transfer. Researchers imaging and collecting data from xenogeneic experiments were unaware of what each treatment group represented, but data were not analyzed in a blinded manner.

### Reporting summary

Further information on research design is available in the Nature Portfolio Reporting Summary linked to this article.

## Data availability

The in vitro and in vivo data generated in this study are provided in the Supplementary Information/Source Data file. Hyperlinks with GenBank accession numbers are provided in the Methods section. Source data are provided with this paper.

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

## Acknowledgements
This work was supported by NIH R01 HL155150 (F.C.), NIH P01 CA111412 (J.S.M.), and research funds provided by Fate Therapeutics.

## Author contributions
F.C., R.B., J.P.G., K.J.M., B.W., U.H., A.R., B.V., and J.S.M. conceptualized the study and developed the methodology. R.B., J.P.G., S.M., S.G., R.A., A.M., K.T., W.W., A.K., B.G., A.W., G.B., J.H., T.D., and T.T.L. performed experiments. F.C., R.B., and J.P.G. analyzed and interpreted data. F.C. drafted the paper. R.B., J.P.G., Z.B.D., and B.V. coordinated and managed experiments. B.V. and J.S.M. supervised the study.

## Competing interests
F.C., K.J.M., and J.S.M. are paid consultants for Fate Therapeutics, and they receive research funds and stock options from this relationship. B.W. receives research funds from Fate Therapeutics. J.S.M. serves on the Scientific Advisory Boards of ONK Therapeutics and Wugan, and is a paid consultant for Vycellix and GT BioPharma (with research funds and stock options) all unrelated to the content of this manuscript. R.B., J.P.L., S.M., S.G., R.A., B.G., A.W., G.B., J.H., T.D., T.T.L., and B.V. are employees of Fate Therapeutics. The remaining authors declare no competing interests. Fate Therapeutics owns patents (METHODS AND COMPOSITIONS FOR INDUCING HEMATOPOIETIC CELL DIFFERENTIATION; Patent No. 10,626,372) covering the iNK cells reported here.
