## [Peer Review File · Nature Communications]

Quadruple gene-engineered natural killer cells enable multi-antigen targeting for durable antitumor activity against multiple myelomaREVIEWER COMMENTS

Reviewer #1 (Remarks to the Author):

Comments to the authors:

Beyond CAR-T cells, treatment concepts using CAR-NK cells are highly promising, especially for treatment of advanced haematological diseases.

The authors are expert in the field of NK cell therapy and here they address if / how iPSC-generated and gene engineered off the shelf NK cell products might efficiently target MM in vitro and in vivo. The concept of iDuo-MM CAR-NK cells combining BCMA-CAR, IL-15/15Ra

The manuscript contains a high quantity of interesting data. Figures are well presented and discussed and the finally drawn conclusions are of important translational interest. However, especially with regard to clinical use, there are concerns which should be further addressed prior to publication.

Major concerns are:

Methods:

- In the figure legends, the authors mention that iDuo-MM CAR-NK cells or expanded PBNK cells were applied immediately thawed from cryopreservation. Was the treatment identical for iNK, iDuo MM CAR NK and PBMC-derived PBNK cells. I am wondering if the PBNK cells would not need a recovery and expansion period prior use in vitro or in vivo as even mentioned by the authors (line 378).
- The authors should provide clear and detailed information on the culturing method used for iDuo CAR NK cells, including data on NK cell viability and expansion rates.
- In the same context, the authors should explain if the PBNK cells also were expanded by co-culture with irradiated K562 cells 468 transduced with membrane-bound IL-21 and 4-1BB ligand.
- Furthermore, freezing and thawing protocols need to be precisely added for both iNK, modified iBK and PBNK, including expansion rates.
- While the authors claim all the needs of NK cells for the gain in optimal cytotoxicity, which has been introduced for the iDuo-MM CAR-NK cells, including IL-15RF, they use unmodified iNK or even PBNK cells as control. In both cases, at least the introduction (or administration during in vitro expansion) of IL-15RF (and not only IL-15) is needed (see also the following comments).
- Which company did provide GSI?
- Which antibodies (clones) have been used for FACS.
- In the material and methods section, CRISPR-mediated target integration is briefly mentioned. The authors should add the details on the method, including the electroporation protocols, kits, instruments used.

Results:

- Fig. 4 in vitro IncuCyte data and Fig 5 in vivo data: the authors compare iDuo MM CAR NK cells with IL-15 expanded PBNKs (immediately after cryopreservation). Indeed, there is a significant advantage in longterm cytotoxicity, but this is not the adequate control. A fair control would be based on the iNK cell product (as in Figure 1, but using iNK cells that also have the IL-15RF and/or the improved CD16).
- Fig. 6 and 7: The dose of CAR T cells and iDuo CAR NK cells significantly differs. The authors explain that GVHD-induction by the CAR-T cell product was the reason for this. However, I believe that further explanation is needed and the authors should provide the GVHD-data of the CAR T cell product as suppl data file from their dose finding experiments.
- Furthermore, only the iDuo CAR NK cells seem to be applied 3 times in contrast to CAR-T cells. If the reason for this is GVHD-induction by CAR-T cells, this should be explained by the persistence of clonally expanding CAR T cells. The authors need to provide data on the persistence of CAR T cells in the FACS analysis of PB.
- In general, I am wondering, why the repetitive application of iDuo CAR NK cells is of need, since these cells are equipped with IL-15RF. The authors should add there migration and in vivo expansion analysis of all different immune cell preparations which have been applied.
- Did the authors measure cytotoxicity against primary MM cells +/- daratumumab in vitro? This would be of high interest concerning clinical translation and should be added.

Discussion:

- Concerning NCT05182073 mentioned in the final discussion, which is based on FT576 (Fate therapeutics): can you give an update on the design, inclusion criteria and current status of the clinical trial, e.g. if already patients have been included.

Reviewer #2 (Remarks to the Author):

This manuscript builds on the previously generated and described iADAPT cells (ref 28) by adding a BCMA CAR into these iPSC-derived NK cells. Treatment of MM cell-line derived tumors with these BCMA CAR iNK cells, combined with daratumumab achieve powerful anti-MM responses in xenograft models. The results are therefore of specific interest for investigators involved in myeloma treatment.

However, I found one important issue which could be of interest for general reader not well studied: Since these iNK cells were generated by knockout of CD38 and additional insertion of a , membrane bound IL-15/IL15R, with the idea to improve their in vivo persistency, it would be relevant to study the in vivo persistence and anti-tumor efficacy also only after a single injection. The authors now show the efficacy after three injections. Since these iNK cells are not devoid of HLA molecules, repeated injections may in the clinical practice induce anti-HLA antibodies or even alloreactive T cells which might limit their in vivo survival and efficacy.

In addition I see that in many experiments expanded PBNK cells were used as controls. In some experiments (at least in fig 4 and 5) it would be also relevant to include the iADAPT NK cells (thus iNK cells without the BCMACAR expression) to compare the efficacy of targeting MM cells with daratumumab only vs BCMACAR only.

From a safety point of view, it would be also relevant to measure the off-tumor on target cytotoxic affects of this new cellular product, when combined with daratumumab. I have not seen in vitro or in vivo data specifically addressing this issue.

There are in addition some (minor) technical, textual issues:

In figure 3: is the caspase 3/7 assay validated to be used as a surrogate of cytotoxicity? measured by a standard cytotoxicity assay measured by chromium release or 7AAD uptake (or another standard cytotoxicity assay based on counting viable cells would be recommended).

In figure 3a it is curious how the "representative of two independent assays" has been chosen.

In figure 3b: it would be relevant to comment on (or even by showing data) why the overnight priming of iDuo-MM-CAR-NK cells with IL-15 is necessary.

In figure 7: GSI+DARA combination is lacking. This control is relevant since GSI not only stabilizes/improves BCMA expression but may also have an important impact on tumor microenvironment induced tumor escape by inhibiting notch-signalling between MM cells and stromal cells.

In the results section, references for figure 2 are wrong (eg: text refers to figure 1c instead of figure 2.)

Reviewer #3 (Remarks to the Author):

The study by Cichocki et al. describes the generation and functional characterization in preclinical models of engineered NK cells for dual targeting of multiple myeloma. These iDuo-MM CAR-NK cells were derived by ex vivo differentiation of CD34+ HSCs, obtained from genetically modified iPSCs that carry a targeted deletion of the CD38 gene and constructs encoding an NK-optimized CAR directed against BCMA, a non-cleavable high-affinity variant of CD16 and a membrane-anchored IL-15 agonist (IL-15RF).

While knockout of CD38 to prevent fratricide, expression of various formats of IL-15 to achieve factor-independence of NK cells, and expression of different CARs or CD16 in iPSC-derived NK cells have previously been described as independent approaches, the novelty of this study is to combine all these strategies in a promising new engineered cell product for targeting of multiple myeloma. The authors convincingly demonstrate that iDuo-MM CAR-NK cells have an NK cell phenotype, efficiently kill BCMA-positive targets through the CAR, display ADCC activity against CD38-positive targets in the presence of daratumumab, and control in vivo growth of MM.1S multiple myeloma cells in a mouse xenograft model largely independent from exogenous cytokines. The data are well presented, and overall the study is very convincing. Nevertheless, the following points should be addressed to further strengthen the paper.

Major:

Typical inhibitory receptors expressed by NK cells (KIRs, CD94/NKG2A) should be included in the phenotypic analysis of iNK and iDuo-MM CAR-NK cells (Fig. 1).

iDuo-MM CAR-NK cells displayed low cytotoxicity to allogeneic PBMCs from healthy donors (Fig. 3B), rated by the authors as 'off-target' cytotoxicity. Did the PBMCs tested contain BCMA-positive B cells possibly triggering the BCMA-CAR?

Do iDuo-MM CAR-NK cells kill normal BCMA-positive B cells in vitro? Potential activity against healthy B cells should also be addressed in the discussion.

In the serial restimulation assays in Fig. 4, the PBNK included for comparison were kept without exogenous cytokine support. Similarly, in the in vivo experiment shown in Fig. 5, PBNK seem to have been applied without co-treatment with IL-2 or IL-15. These experiments nicely demonstrate the expected independence of iDuo-MM CAR-NK cells from exogenous cytokines in contrast to PBNK. Nevertheless, to provide a fair comparison, at least in the in vitro experiments serial restimulation of PBNK should also be performed in the presence of exogenous IL-2 or IL-15 as it would be done for donor NK transfer in a clinical setting.

Did iDuo-MM CAR-NK cells and PBNK upregulate exhaustion markers upon repeated stimulation?

Is there a bystander effect of iDuo-MM CAR-NK cells through their IL-15RF towards PBNK that would support PBNK growth or functionality?

When comparing to the data from rounds 1, 2 and 3 in Fig. 4A and B, it appears that quantification of remaining MM.1S cells after serial restimulation of effector cells shown in Fig. 4C and D was performed after round 1. This should be clarified.

How does in vivo persistence and expansion of iDuo-MM CAR-NK cells (Fig. 5C) compare to that of BMCA-CAR T cells?

If available, survival data should be included for the in vivo experiments shown in Fig. 5, 6 and 7.

Minor:

It is mentioned that the iDuo-MM CAR-NK cells tested were of clonal origin, but it remains unclear whether this relates to the origin of the iPSCs or the final NK cell product. Are all iDuo-MM CAR-NK cells of a certain batch of single-cell origin with the same number and positions of CAR vector integration or polyclonal with respect to the CAR? This should be clarified.

The env pseudotype used for the lentiviral CAR vector should be indicated.

The text in line 169 refers to Fig. 2, not as wrongly stated to Fig. 1C.

Fig. 1C appears to show one example each for unmodified iNK and iDuo-MM CAR-NK cells, while the respective legend mentions analysis of several independent iPSC clones. This should be made consistent.

In Fig. 4 the labeling of panels (E) and (D) should be exchanged to be in line with the description in the figure legend.

In Fig. 5C a ' μ ' seems to be missing from 'human NK cells/ I blood'.

Point-by-point reply for NCOMMS-22-07254-T “Multi-Antigen Targeting, Off-the-Shelf CAR-NK cells Demonstrate Durable Efficacy in Multiple Myeloma

We thank all 3 reviewers for carefully reading this manuscript and making suggestions to improve the overall its overall strength. We have performed multiple additional experiments to address major comments provided by the reviewers and have added additional data that was requested. We believe that we have substantially improved the manuscript by addressing these comments and making the appropriate changes in the main text. Below is a point-by-point reply to the reviewers (replies in blue text):

Reviewer 1:

Major comments:

Methods:

1. In the figure legends, the authors mention that iDuo-MM CAR-NK cells or expanded PBNK cells were applied immediately thawed from cryopreservation. Was the treatment identical for iNK, iDuo-MM CAR-NK and PBMC-derived PBNK cells? I am wondering if the PBNK cells would not need a recovery and expansion period prior to use in vitro or in vivo as even mentioned by the authors (line 378).

Yes. iDuo-MM CAR-NK cells, backbone iNK cells, and PBMC-derived PBNK cells were all immediately thawed from cryopreservation and used directly in functional experiments. In response to the reviewer, we present the above data showing results from a specific cytotoxicity assay measuring killing against K562 cells at multiple E:T ratios. PBNK cells from 2 healthy donors were expanded for 2 weeks on K562 feeders engineered to express mbIL-21 and 4-1BBL. The NK cells were then cryopreserved and thawed immediately prior to the functional assays. In our hands, PBNK cell cytotoxicity is well preserved post-thaw without the need for additional stimulation or recovery.

2. The authors should provide clear and detailed information on the culturing method used for iDuo-MM CAR-NK cells, including data on NK cell viability and expansion rates.

Additional information on the generation of iDuo-MM CAR-NK cells has been added to the Materials and Methods section, and data on viability and expansion rates have been included in Supplemental Figure 2.

3. In the same context, the authors should explain if the PBNK cells were also expanded by co-culture with irradiated K562 cells transduced with membrane-bound IL-21 and 4-1BB ligand.

Yes. The PBNK cells were expanded with the same feeder cells as the iNK cells were and for the same length of time. This information has been added to the Methods section.

4. Furthermore, freezing and thawing protocols need to be precisely added for iNK cells, modified iNK cells and PBNK cells, including expansion rates.

Additional information on the thawing protocol has been included in the Materials and Methods section. Expansion rates for iDuo-MM CAR-NK cells has been added as Supplemental Figure 2.

5. While the authors claim all the needs of NK cells for the gain in optimal cytotoxicity, which has been introduced for the iDuo-MM CAR-NK cells, including IL-15RF, they use unmodified iNK or even PBNK cells as controls. In both cases, at least the introduction (or administration during in vitro expansion) of IL-15RF (and not only IL-15) is needed (see also the following comments).

-Which company did provide GSI?

-Which antibodies (clones) have been used in FACS?

We appreciate the first point made by the reviewer about having additional controls/comparisons in both in vitro and in vivo experiments testing the antitumor function of iDuo-MM CAR-NK cells. To address this point, we included data from additional IncuCyte restimulation experiments comparing MM.1R cells alone to backbone iNK cells (hnCD16/CD38 knockout/IL-15RF) and iDuo-MM CAR-NK cells (hnCD16/CD38 knockout/IL-15RF/anti-BCMA CAR) in the presence or absence of daratumumab at a 3:1 E:T ratio. Similar to experiments performed with PBNK cells, backbone iNK cells demonstrated relatively high cytotoxicity early in the assay (particularly when combined with daratumumab) but failed to kill MM targets at later time points in the experiment. This data is included as Figure 4F, G in the revised manuscript. We also provide additional in vivo data (Figure 5A, E, F, G in the revised manuscript) comparing multi-dose administration of backbone iNK cells, iDuo-MM CAR-NK cells and expanded PBNK cells in the MM.1S-luc model. Similar to the in vitro experiments in Figure 4, backbone NK cells were unable to effectively control tumor burdens despite survival support from IL-15RF and high in vivo persistence. Together, this new data shows the importance of IL-15RF for in vivo persistence and BCMA targeting by the CAR for sustained tumor control. Additional information on the source of GSI and antibody clones used in FACS have been added to the Methods section.

6. In the material and methods section, CRISPR-mediated target integration is briefly mentioned. The authors should add the details on the method, including electroporation protocols, kits, instruments used.

Additional information on CRISPR-mediated integration, electroporation, kits, and instruments is included in the Materials and Methods section.

Results:

1. Fig. 4 in vitro IncuCyte data and Fig. 5 in vivo data: the authors compare iDuo-MM CAR-NK cells with IL-15-expanded PBNKs (immediately after cryopreservation). Indeed, there is a significant advantage in long-term cytotoxicity, but this is not the adequate control. A fair control would be based on the iNK cell product (as in Figure 1, but using iNK cells that also have the IL-15RF and/or the improved CD16).

We agree with the point made by the reviewer, and we have added additional data to Figures 4 and 5. See the response to point # 5 in the above *Methods* section.

2. Fig. 6 and 7: the dose of CAR-T cells and iDuo-MM CAR-NK cells significantly differs. The authors explain that GvHD-induction by the CAR-T cell product was the reason for this. However, I believe that further explanation is needed, and the authors should provide the GvHD data of the CAR-T cell product as a supplemental data file from their dose finding experiments. Furthermore, only the iDuo-MM CAR-NK cells seem to be applied 3 times in contrast to CAR-T cells. If the reason for this is GvHD induction by CAR-T cells, this should be explained by the persistence of clonally expanding CAR T cells. The authors need to provide data on the persistence of CAR-T cells in the FACS analysis of peripheral blood.

Our mention that we used a single dose of CAR-T cells at a lower dose to avoid GvHD was based on previous work with adoptive transfer of activated T cells into tumor-bearing NSG mice where we observed evidence of GvHD 4-5 weeks after transfer (Cichocki et al., *Sci. Transl. Med.*, 2020). Because of that previous experience we used a lower dose of anti-BCMA CAR-T cells in the current study. We did not observe signs of GvHD in the experiments shown in Figures 6 and 7. To the second point made by the reviewer, we added peripheral blood counts for all iDuo-MM CAR-NK cell and anti-BCMA CAR-T cell conditions in Figure 7 (new Figure 7F). We did not observe differences in persistence/expansion between the two cell types in peripheral blood. Overall, cell counts were relatively low. However, in these experiments the numbers of adoptively transferred iDuo-MM CAR-NK cells were reduced to assess the combined effects of daratumumab and GSI on tumor control in vivo.

3. In general, I am wondering why repetitive application of iDuo-MM CAR-NK cells is needed since these cells are equipped with IL-15RF. The authors should add their migration and in vivo expansion analysis of all different immune cell preparations which have been applied.

The reviewer raises a fair point about single vs. multiple dose administrations of iNK cells. To address this comment, we included additional data in Figure 5 where groups of mice were administered one dose of 1×10^7 iDuo-MM CAR-NK cells alone or in combination with daratumumab. Tumor burden over time in these mice was compared to MM.1S tumor alone and

MM.1S tumor plus daratumumab. Our data shows that single agent treatment of either daratumumab or iDuo-MM CAR-NK cells resulted in moderate reductions in tumor burden over time, while the combination of iDuo-MM CAR-NK cells resulted in robust and durable tumor control. We also measured persistence of iDuo-MM CAR-NK cells +/- daratumumab after single dose administration and saw moderate cell numbers at day 22 that declined substantially during subsequent weeks. Thus, after single dose administration, iDuo-MM CAR-NK cells were effective when combined with daratumumab but did not persist at high numbers in the blood. We compared these results with mice that received 3 doses of either iDuo-MM CAR-NK cells, expanded PBNK cells or backbone iNK cells. In this model, 3 doses of iDuo-MM CAR-NK cells led to significant tumor control, while 3 doses of either expanded PBNK cells or backbone iNK cells was ineffective. Consistent with our previous study (Woan et al., *Cell Stem Cell*, 2021), backbone iNK cells (termed iADAPT in our previous study) did not control tumor burden in vivo despite high levels of persistence in the blood that was comparable to that of iDuo-MM CAR-NK cells. This data shows the importance of activation through the anti-BCMA CAR for in vivo antitumor function. Thus, multi-dose administration is important for adoptive therapy with iDuo-MM CAR-NK cells alone and for high persistence of the cells in the peripheral blood, but single doses of these cells can be effective when combined with therapeutic monoclonal antibodies. The new data described above has been added to Figure 5.

4. Did the authors measure cytotoxicity against primary MM cells +/- daratumumab in vitro? This would be of high interest concerning clinical translation and should be added.

To address the reviewer's comment and elevate the clinical significance of this study, we obtained bone marrow samples from 2 patients with relapsed MM. CD138⁺ MM cells were enriched from the marrow samples and co-cultured with iDuo-MM CAR-NK cells at a 2:1 E:T ratio to assess cytotoxic responses by flow cytometry. For patient 1, we had enough cells to also include daratumumab conditions. We observed substantial reductions in the MM population after co-culture with iDuo-MM CAR-NK cells that was further reduced with the addition of daratumumab. We had limited material for patient 2, which precluded our ability to add daratumumab conditions. However, the MM cell population from the marrow of this patient also decreased markedly upon co-culture with iDuo-MM CAR-NK cells, with a near eradication of MM cells. This new data has been added as Figure 3D.

Discussion:

1. Concerning NCT05182073 mentioned in the final discussion, which is based on FT576 (Fate Therapeutics): can you give an update on the design, inclusion criteria and current status of the clinical trial, e.g. if patients have already been included?

We have added a sentence at the end of the discussion mentioning the current state of the clinical trial.

Reviewer 2:

Major comments:

1. I found one important issue which could be of interest for the general reader that was not well studied: Since these iNK cells were generated by knockout of CD38 and additional insertion of a membrane-bound IL-15/IL-15R, with the ideal to improve their in vivo persistence, it would be relevant to study the in vivo persistence and anti-tumor efficacy also only after a single injection. The authors now show the efficacy after three injections. Since these iNK cells are not devoid of HLA molecules, repeated injections may in the clinical practice induce anti-HLA antibodies or even alloreactive T cells that might limit their in vivo survival and efficacy.

The reviewer raises a fair point about single vs. multiple dose administrations of iNK cells. This was also brought up by Reviewer 1. To address this comment, we included additional data in Figure 5 where groups of mice were administered one dose of 1×10^7 iDuo-MM CAR-NK cells alone or in combination with daratumumab. Tumor burden over time in these mice was compared to MM.1S tumor alone and MM.1S tumor plus daratumumab. Our data shows that single agent treatment of either daratumumab or iDuo-MM CAR-NK cells resulted in moderate reductions in tumor burden over time, while the combination of iDuo-MM CAR-NK cells resulted in robust and durable tumor control. We also measured persistence of iDuo-MM CAR-NK cells +/- daratumumab after single dose administration and saw moderate cell numbers at day 22 that declined substantially during subsequent weeks. Thus, after single dose administration, iDuo-MM CAR-NK cells were effective when combined with daratumumab but did not persist at high numbers in the blood. We compared these results with mice that received 3 doses of either iDuo-MM CAR-NK cells, expanded PBNK cells or backbone iNK cells. In this model, 3 doses of iDuo-MM CAR-NK cells led to significant tumor control, while 3 doses of either expanded PBNK cells or backbone iNK cells was ineffective. Consistent with our previous study (Woan et al., *Cell Stem Cell*, 2021), backbone iNK cells (termed iADAPT in our previous study) did not control tumor burden in vivo despite high levels of persistence in the blood that was comparable to that of iDuo-MM CAR-NK cells. This data shows the importance of activation through the anti-BCMA CAR for in vivo antitumor function. Thus, multi-dose administration is important for adoptive therapy with iDuo-MM CAR-NK cells alone and for high persistence of the cells in the peripheral blood, but single doses of these cells can be effective when combined with therapeutic monoclonal antibodies. The new data described above has been added to Figure 5.

2. In addition, I see that in many experiments, expanded PBNK cells were used as controls. In some experiments (at least in Fig. 4 and Fig. 5) it would be relevant to include the iADAPT NK cells (thus NK cells without the BCMA CAR expression) to compare the efficacy of targeting MM cells with daratumumab only vs. BCMA CAR only.

The response to the in vivo aspect of this comment (Fig. 5) can be seen in the response above where data showing antitumor responses and in vivo persistence of backbone iNK cells (iADAPT NK) were included. To address the in vitro aspect of this comment, we included data from additional IncuCyte restimulation experiments comparing MM.1R cells alone to backbone iNK cells (iADAPT) and iDuo-MM CAR-NK cells in the presence or absence of daratumumab at a 3:1 E:T ratio. Similar to experiments performed with PBNK cells, backbone iNK cells demonstrated relatively high cytotoxicity early in the assay (particularly when combined with

daratumumab) but failed to kill MM targets at later time points in the experiment. This data is included as Figure 4F, G in the revised manuscript.

3. From a safety point of view, it would also be relevant to measure the off-tumor on-target cytotoxic effects of this new cellular product when combined with daratumumab. I have not seen in vitro or in vivo data specifically addressing this issue.

BCMA expression is generally restricted to plasma cells. Unfortunately, direct analysis of healthy plasma cells in peripheral blood by flow cytometry is hampered by their extremely low frequency in this tissue (doi:10.1046/j.1365-2249.2002.2025.x.). For this reason, healthy bone marrow or lymph tissue (which we do not currently have access to) has historically been used to isolate and study plasma cells. We could not reliably assess potential iDuo-MM CAR-NK cell cytotoxicity against plasma cells in healthy peripheral blood because of the low frequency of this population. We added text to the Discussion section of the manuscript to acknowledge the possibility of iDuo-MM CAR-NK cell cytotoxicity against healthy plasma cells after adoptive transfer into MM patients.

Minor comments:

1. In Figure 3: is the caspase 3/7 assay validated to be used as a surrogate of cytotoxicity? Another cytotoxicity assay measured by chromium release or 7AAD uptake (or another standard cytotoxicity assay based on counting viable cells) would be recommended. In Fig. 3A, it is curious how “representative of two independent assays” has been chosen.

The assay that we used in Figure 3 for assessment of cytotoxicity based on cleaved caspase 3 in target cells has previously been validated thoroughly for assessment of cytotoxic T cell killing and shown to yield comparable results as ⁵¹CR-release but with higher sensitivity and to be accurate at both high and low effector cell frequencies (He et al., *J. Immunol. Methods.*, 2005). Based on this study and extensive use by our own laboratories, we believe that detection of caspase 3 cleavage is an accurate surrogate for specific cytotoxicity. More information on calculations used for the assay have been added to the Methods section. In Figure 3A, the data was generated from 2 independent experiments (backbone clone #1 and iDuo-MM CAR-NK cells Batch #1 in the first experiment and backbone clone #2 and iDuo-MM CAR-NK cells Batch #2 in the second experiment).

2. In Figure 3B it would be relevant to comment on (or even show data) why the overnight priming of iDuo-MM CAR-NK cells with IL-15 is necessary.

In Figure 3B, the PBNK cells were primed overnight with IL-15 (to mimic conditions used historically in adoptive NK cell transfer clinical studies), but the iDuo-MM CAR-NK cells that were used were thawed immediately from cryopreservation and added to targets. We apologize if there was a misunderstanding here.

3. In Figure 7 GSI + daratumumab combination is lacking. This control is relevant since GSI not only stabilizes/improves BCMA expression but may also have an important

impact on tumor microenvironment induced tumor escape by inhibiting notch signaling between MM cells and stromal cells.

We thank the reviewer for the suggestion. Data showing this control condition has been added to Figure 7.

4. In the results section, references for Figure 2 are wrong (e.g. text refers to Figure 1C instead of Figure 2).

This error has been fixed.

Reviewer 3:

Major comments:

1. Typical inhibitory receptors by NK cells should be included in the phenotypic analysis of iNK and iDuo-MM CAR-NK cells (Fig. 1).

We have added additional data to Figure 1B showing surface level frequencies of KIR2DL1 and KIR2DL2/3 on freshly isolated PBNK cells, expanded PBNK cells, unmodified iNK cells, and iDuo-MM CAR-NK cells.

2. iDuo-MM CAR-NK cells displayed low cytotoxicity to allogeneic PBMCs from healthy donors (Fig. 3B), rated by the authors as “off-target” cytotoxicity. Did the PBMCs tested contain BCMA-positive B cells possibly triggering the BCMA-CAR?

BCMA expression is generally restricted to plasma cells. Unfortunately, direct analysis of healthy plasma cells in peripheral blood by flow cytometry is hampered by their extremely low frequency in this tissue (doi:10.1046/j.1365-2249.2002.2025.x.). For this reason, healthy bone marrow or lymph tissue (which we do not currently have access to) has historically been used to isolate and study plasma cells. We could not reliably assess potential iDuo-MM CAR-NK cell cytotoxicity against plasma cells in healthy peripheral blood because of the low frequency of this population.

3. Do iDuo-MM CAR-NK cells kill normal BCMA-positive B cells in vitro? Potential activity against healthy B cells should also be addressed in the discussion.

As mentioned above, the lack of plasma cells in healthy blood precluded our ability to assess killing in vitro. We added text to the Discussion section of the manuscript to acknowledge the possibility of iDuo-MM CAR-NK cell cytotoxicity against healthy plasma cells after adoptive transfer into MM patients.

4. In the serial restimulation assays in Fig. 4, the PBNK included for comparison were kept without exogenous cytokine support. Similarly, in the in vivo experiment shown in Fig. 5, PBNK seem to have been applied without co-treatment with IL-2 or IL-15. These experiments nicely demonstrate the expected independence of iDuo-MM CAR-NK cells

from exogenous cytokines in contrast to PBNK. Nevertheless, to provide a fair comparison, at least in the in vitro experiments serial restimulation of PBNK should also be performed in the presence of exogenous IL-2 or IL-15 as would be done for donor NK cell transfer in a clinical setting.

We acknowledge the reviewer's point that for a fair comparison to be made, data from in vitro functional assays should be included comparing iDuo-MM CAR-NK cells to another NK cell product that has cytokine support. To address this point, we included data from additional IncuCyte restimulation experiments comparing MM.1R cells alone to backbone iNK cells (hnCD16/CD38 knockout/IL-15RF) and iDuo-MM CAR-NK cells (hnCD16/CD38 knockout/IL-15RF/anti-BCMA CAR) in the presence or absence of daratumumab at a 3:1 E:T ratio. Similar to experiments performed with PBNK cells, backbone iNK cells demonstrated relatively high cytotoxicity early in the assay (particularly when combined with daratumumab) but failed to kill MM targets at later time points in the experiment. This data is included as Figure 4F, G in the revised manuscript. We also provide additional in vivo data (Figure 5A, E, F, G in the revised manuscript) comparing multi-dose administration of backbone iNK cells, iDuo-MM CAR-NK cells and expanded PBNK cells in the MM.1S-luc model. Similar to the in vitro experiments in Figure 4, backbone NK cells were unable to effectively control tumor burdens despite survival support from IL-15RF and high in vivo persistence. We took this approach to including additional conditions with cytokine support (instead of adding exogenous cytokines to various conditions in repeat experiments) to also address similar comments made by Reviewer 1. Together, this new data shows the importance of IL-15RF for in vivo persistence and BCMA targeting by the CAR for sustained tumor control.

5. Did iDuo-MM CAR-NK cells and PBNK cells upregulate exhaustion markers upon repeated stimulation:

We provide additional data (Supplementary Figure 1) showing the frequencies of LAG-3 and TIM-3 on expanded PBNK cells and iDuo-MM CAR-NK cells at the end of serial restimulation experiments (Figure 4A). This data shows a small upregulation of LAG-3 on PBNK cells that is further increased with the addition of daratumumab without noticeable expression on iDuo-MM CAR-NK cells. TIM-3 was observed at higher frequencies of both PBNK cells and iDuo-MM CAR-NK cells and was modestly elevated with the addition of daratumumab.

6. Is there a bystander effect of iDuo-MM CAR-NK cells through their IL-15RF towards PBNK cells that would support their growth or functionality?

We have performed co-culture experiments with allogeneic peripheral blood CD8⁺ T cells and backbone iNK cells and non-transduced iNK cells previously where the CD8⁺ T cells were pre-labeled with CFSE dye. We did not observe any significant proliferation of T cells in these co-cultures, while T cells cultured at the same time with exogenous IL-15 underwent multiple cell divisions. The same experimental set up was attempted with peripheral blood NK cells, but they were much more difficult to distinguish from iNK cells given their phenotypic similarities. Since this data is still preliminary (and is ancillary to the main messages of the manuscript), we decided not to publish these results at this time.

7. When comparing to the data from rounds 1, 2 and 3 in Fig. 4A and B, it appears that quantification of remaining MM.1S cells after serial restimulation of effector cells shown in Fig. 4C and D was performed after round 1. This should be clarified.

Quantification of remaining MM.1S cells and effector cells was performed at the end of the assay (after round 3). This has been clarified in the labeling of Figures 4B-E and G.

8. How does in vivo persistence and expansion of iDuo-MM CAR-NK cells compare to that of BCMA-CAR T cells?

We added peripheral blood counts for all iDuo-MM CAR-NK cell and anti-BCMA CAR-T cell conditions in Figure 7 (new Figure 7F). We did not observe differences in persistence/expansion between the two cell types in peripheral blood. Overall, cell counts were relatively low. However, in these experiments the numbers of adoptively transferred iDuo-MM CAR-NK cells were reduced to assess the combined effects of daratumumab and GSI on tumor control in vivo.

9. If available, survival data should be included for the in vivo experiments shown in Fig. 5, 6 and 7.

Unfortunately, mice from these experiments were not monitored long term for survival.

Minor comments:

1. It is mentioned that the iDuo-MM CAR-NK cells tested were of clonal origin, but it remains unclear whether this relates to the origin of the iPSCs or the final NK cell product. Are all iDuo-MM CAR-NK cells of a certain batch of single-cell origin with the same number and positions of CAR vector integration or polyclonal with respect to the CAR? This should be clarified.

All iDuo-MM CAR-NK cells were from single-cell iPSC origin with clonal expression of the anti-BCMA CAR. This has been clarified in the methods section.

2. The env pseudotype used for the lentiviral CAR vector should be indicated.

The lentiviral CAR vector was a VSV-G pseudotype. This information has been added to the Methods section.

3. The text in line 169 refers to Fig. 2, not as wrongly stated to Fig 1C.

This error has been fixed.

4. Fig 1.C appears to show one example each for unmodified iNK and iDuo-MM CAR-NK cells, while the respective legend mentions analysis of several independent iPSC clones. This should be made consistent.

There were errors in the legend for Figure 1C. This legend has now been corrected.

5. In Fig. 4 the labeling of panels (E) and (D) should be exchanged to be in line with the description in the figure legend.

The figure legend has been fixed to accurately describe each subfigure.

6. In Fig. 5C a μ seems to be missing from 'human NK cells/ l blood'.

The current label in this figure is correct.

REVIEWER COMMENTS

Reviewer #1 (Remarks to the Author):

The authors successfully addressed all minor and major comments. The revised version of the manuscript has been significantly improved.

Reviewer #2 (Remarks to the Author):

I have carefully read the response of authors to all reviewers' comments and the revised paper. I appreciate the additional information provided. However in the light of these new data there are still important issues.

1. The new data provided in figure 5 now shows clearly that after single dose administration iDuo-MM CAR-NK cells alone were modestly effective as they did not not persist at high numbers in the blood.

However, the abstract still reads (line 52/35): “we introduced a membrane-bound IL-15/IL-15R fusion molecule to enhance function and persistence” The paper claim this in the discussion as well. However this in the light of data in figure 5 this is a clear overstatement. The manuscript shows nowhere any additional value of IL15/IL15R fusion molecule neither for persistence nor for the activity of the iDuo-MM CAR-NK cells.

2. The new experiments in figure 5 reveals another important issue: The BCMA targeting via iDuo-MM CAR-NK may be necessary , but it is probably not effective enough, because the iDuo-MM CAR-NK cells need either to be combined with Daratumumab or they need to be injected repeatedly.

In my original review I had already mentioned the possibility that repeated injections in the clinical setting could be problematic due to induction of anti HLA antibodies (This question is not answered).

The problem of combination studies with Daratumumab is that the figure 5 (or any other figure) does not show any data whether Daratumumab targeting only would also be similarly effective because the authors left out the control which would be the combination of iNK (iADAPT NK) cells with Daratumumab, thus CD38 targeting without BCMA targeting. This was actually shown in their previous Cell Stem Cell 28, 2062–2075, December 2, 2021 paper. When I compare the results shown in that paper I already have the impression that at least or above 80% of the anti-tumor effect seen in MM1S model is achieved only by Daratumumab targeting, thus without BCMA targeting.

In short, the authors cannot convincingly show that additional BCMA targeting by iDuo-MM CAR-NK cells have an additional value above iADAPT NK cells if both products are also targeting CD38 by the addition of Daratumumab. This hampers the novelty and superiority of this new product.

3. Newly provided in vitro data using primary MM cells in figure 3 further points to possible limitations of BCMA targeting only: The authors clearly mention that one of the primary MM samples could be effectively eliminated only if iDuo-MM CAR-NK cells were combined with Daratumumab. This suggests that BCMA expression on these MM cells is probably not sufficient to be effectively targeted by BCMA CAR. Unfortunately however, BCMA expression on the MM cells is not determined. Thus the reader is not able to judge if a low

BCMA+ MM cells can also be targeted by iDuo-MM CAR-NK cells. When seeing this heterogeneity, the authors should have actually tested more primary MM samples and compared the MM lysis efficacy with BCMA expression levels. Again in this figure Daratumumab only (in the presence of iADAPT cells) is missing. Thus again the reader is not able to estimate the impact of CD38 targeting only.

May be because of this the clinical trial has now been started not with iDuo-MM CAR-NK only but in combination with Daratumumab.

However combination of these cells with daratumumab has some potential issues Therefore, 4. In my original review I asked to provide the reader with data on off tumor on target toxicity. especially when iDuo-MM CAR-NK cells are combined with Daratumumab. The authors discuss about the impossibility to do experiments with healthy plasma cells, but my question was actually more about several healthy hematopoietic cells that express CD38, all of which will be targeted by Daratumumab. These CD38+ cells are readily available in BM. Even without separating them from other cells it is possible to assess their lysis, subset by subset, using FACS-based cytotoxicity assays. Thus the potential toxicity of iDuo-MM CAR-NK cells when they are combined with Daratumumab is still missing.

5. Finally as a minor important point, I asked to combine GSI only with Daratumumab (of course in the presence of BCMA-CAR negative iNK (iADAPT NK cells) I still do not see the answer to my question.

Reviewer #3 (Remarks to the Author):

New data and additional information have been added to the revised manuscript, adequately addressing my prior questions and comments.

Please correct the following minor inconsistencies:

Most likely by error, the legend of Figure 4 (F) mentions in addition to Backbone iNK and iDuo-MM CAR-NK also serial restimulation of expanded PBNK, without such data shown in the Figure or mentioned in the Results.

For enhanced clarity, the experiments shown in Figure 7, panels A and B versus panels C to F should be described separately in the legend. For A and B, it should be indicated that NSG mice were first inoculated with MM.1S cells before investigating BCMA surface expression upon treatment with LY 3039478. The number of NSG mice mentioned (38) appears to refer to the separate adoptive therapy experiments shown in C to F, not to A and B.

Point-by-point response for NCOMS-22-07254A “Multi-Antigen Targeting, Off-the-Shelf CAR-NK cells Demonstrate Durable Efficacy in Multiple Myeloma”

We thank the reviewers and editor for their additional comments and suggestions. Our responses can be seen below in blue text:

Reviewer #2:

1. The new data provided in figure 5 now shows clearly that after single dose administration, iDuo-MM CAR-NK cells alone were modestly effective as they did not persist at high numbers in the blood. However, the abstract still reads (line 52/35): “we introduced a membrane-bound IL-15/IL-15R fusion molecule to enhance function and persistence”. The paper claims this in the discussion as well. However, this in the light of data in figure 5 is a clear overstatement. The manuscript shows nowhere any additional value of IL-15/IL-15R fusion molecule neither for persistence nor for the activity of the iDuo-MM CAR-NK cells.

We acknowledge the point made by the reviewer that iDuo-MM CAR-NK cells (which express IL-15RF) exhibit much lower persistence in the peripheral blood compared to the levels of persistence that are seen with a multi-dose strategy (3 doses of cells). We have added text to the Discussion section of the manuscript to acknowledge this caveat (lines 465-470). However, we strongly disagree with the conclusion that iDuo-MM CAR-NK cells alone were modestly effective and that we provide no evidence for the value of IL-15RF in supporting in vivo persistence of adoptively transferred iDuo-MM CAR-NK cells. In regard to antitumor activity, in Figure 5 (revised, please see response to question 3), we clearly show that one dose of iDuo-MM CAR-NK cells exhibit significant antitumor effect compared to the MM.1S only control arm, the single or multiple dose arms of backbone iNK cell that are not targeted to the tumor, and the PBNK cell control arm that is dosed 3 times. We also show the antitumor activity of iDuo-MM CAR-NK cells is further enhanced when dosed 3 times. In regard to persistence, in Figure 5F, we show that with 3 doses of effector cells, backbone (hnCD16/CD38KO/ IL-15RF) iNK cells or iDuo-MM CAR-NK cells (same backbone modifications plus the anti-BCMA CAR) persist at very high levels (approximately 100 cells per μ L of blood) in the peripheral blood of mice in the absent of exogenous cytokine support, while at the same timepoints PBNK cells display only moderate persistence (approximately 1 cell per μ L of blood). However as discussed above, because the iDuo-MM CAR-NK cells are directed to the MM.1S tumor cells, they significantly reduce tumor burden in this model, while backbone iNK cells do not (because of the lack of antigen targeting). The in vivo antitumor function of iDuo-MM CAR-NK cells is also clearly demonstrated in Figures 6 and 7.

As mentioned in the text added to the Discussion, it is possible that, in the single dose model, a substantial fraction of the cells injected into the mice leave the peripheral blood and traffic to other tissues (spleen, etc.). Multiple doses may be required to support sufficiently high persistence of the cells within circulating peripheral blood. We are actively exploring the reasons for these findings.

2. The new experiments in figure 5 reveal another important issue: The BCMA targeting via iDuo-MM CAR-NK may be necessary, but it is probably not effective enough because the iDuo-MM CAR-NK cells need either to be combined with Daratumumab or they need to be injected repeatedly. In my original review, I had mentioned the possibility that repeated injections in the clinical setting could be problematic due to induction of anti-HLA antibodies (this question is not answered).

While we clearly show that iDuo-MM CAR-NK cells have significant antitumor activity, we agree that multiple dosing and/or the combination with daratumumab improves antitumor activity. In fact, we show that our combination strategy results in antitumor activity that is superior to primary CAR T cells, which is a remarkable achievement and a major point of this manuscript.

In regard to anti-HLA antibody response from repeat dosing of adoptive cell therapy, there is the potential for the emergence of anti-HLA antibodies in patients who receive an allogeneic cell product (be it iNK cells or any other cell type). Several groups, in addition to us, are actively engaged in clinical trials that include multi-dosing of cell therapy. While the data is preliminary, to our knowledge, clear evidence of a humoral response has not been observed.

The problem of combination studies with daratumumab is that the figure 5 (or any other figure) does not show any data whether daratumumab targeting only would also be similarly effective because the authors left out the control which would be the combination of iNK (iADAPT NK) cells with daratumumab, thus CD38 targeting without BCMA targeting. This was actually shown in their previous *Cell Stem Cell* paper. When I compare the results shown in that paper, I already have the impression that at least or above 80% of the anti-tumor effect seen in the MM.1S model is achieved only by daratumumab targeting, thus without BCMA targeting. In short, the authors cannot convincingly show that additional BCMA targeting by iDuo-MM CAR-NK cells have an additional value above iADAPT NK cells if both products are also targeting CD38 by the addition of daratumumab. This hampers the novelty and superiority of this new product.

To address the reviewer's critiques, we have added additional data to Figure 5A and B. We included BLI data showing mice treated with a single dose of backbone iNK (iADAPT NK) cells alone and in combination with daratumumab. This data again shows that backbone iNK cells are unable to control MM.1S tumor growth in vivo but can control tumor growth when combined with daratumumab. The sustained control of tumor burden by iDuo-MM CAR-NK cells combined with daratumumab is significantly greater than that of backbone iNK cells with daratumumab. These data demonstrate the additional benefit of dual targeting through the anti-BCMA CAR. We did not include these data initially, because we believed that we had already made this point with the in vitro experiments included in Figure 4F. However, we have added this new data to confirm importance of dual targeting in vivo.

3. Newly provided in vitro data using primary MM cells in figure 3 further points to possible limitations of BCMA targeting only. The authors clearly mention that one of the primary MM samples could be effectively eliminated only if iDuo-MM CAR-NK cells were combined with daratumumab. This suggests that BCMA expression on these MM cells is not sufficient to be effectively targeted by BCMA CAR. However, BCMA expression on the MM cells is not determined. Thus, the reader is not able to judge if low

BCMA⁺ MM cells can also be targeted by iDuo-MM CAR-NK cells. When seeing this heterogeneity, the authors should have actually tested more primary MM samples and compared the MM lysis efficacy with BCMA expression levels. Again, in this figure daratumumab only (in the presence of iADAPT cells) is missing. Thus, the reader is not able to estimate the impact of CD38 targeting only. May be because of this the clinical trial has now been started not with iDuo-MM CAR-NK cells only but in combination with daratumumab.

We acknowledge the reviewer's points about potential heterogeneity in BCMA expression on MM cells within the patient population. Unfortunately, the numbers of MM cells available from biopsies for research purposes are often very low. Also, patient cells need to be used fresh since they do not survive the freeze/thaw process well. This limits the scope of the experiments that we can do. The purpose of the experiments included in the manuscript was to demonstrate the efficacy of iDuo-MM CAR-NK cells in killing primary MM cells in the presence and absence of daratumumab. This was acknowledged by the editor, and no further experiments were requested. However, we have added text to the Discussion section (lines 452-458) acknowledging that the potential heterogeneity in BCMA levels on MM cells from different individuals could impact the efficacy of iDuo-MM CAR-NK cells as a monotherapy.

4. In my original review I asked to provide the reader with data on off tumor on target toxicity, especially when iDuo-MM CAR-NK cells are combined with daratumumab. The authors discuss the impossibility to do experiments with healthy plasma cells, but my question was actually more about several healthy hematopoietic cells that express CD38, all of which will be targeted by daratumumab. These CD38⁺ cells are readily available in BM. Even without separating them from other cells, it is possible to assess their lysis, subset by subset, using FACS-based assays. Thus, the potential toxicity of iDuo-MM CAR-NK cells when they are combined with daratumumab is still missing.

We agree with the reviewer that CD38⁺ HPCs are present in the bone marrow. However, setting up a functional assay with unsorted cells from bone marrow would likely not yield meaningful data if CD38⁺ HPCs comprise a minor fraction of all cells within the tissue. We are also very limited with respect to healthy bone marrow that could potentially be used for these assays. To address the reviewer's critique, we isolated CD34⁺ HPCs from umbilical cord blood. These cells had uniformly high expression of CD38. We performed 5-hour in vitro co-culture assays with iDuo-MM CAR-NK cells as effectors and dye-labeled CD34⁺CD38⁺ HPCs as targets at a 0.5:1 E:T ratio in the presence or absence of daratumumab. We selected a E:T ratio to be within range of a physiological setting. We had to select only 1 E:T ratio because of limited numbers of CD34⁺CD38⁺ HPCs that could be isolated from each cord blood product. No evidence of cytotoxicity was observed when iDuo-MM CAR-NK cells were cultured with CD34⁺CD38⁺ HPCs alone. With the addition of daratumumab, a small but statistically insignificant decrease in HPC viability was observed. This data has been added as the new Supplementary Figure 1.

5. Finally, as a minor point, I asked to combine GSI only with daratumumab (of course in the presence of BCMA CAR-negative iNK cells. I still do not see the answer to my question.

The “of course in the presence of BCMA CAR-negative iNK cells” part was not clear in the previous critique. That is why we added the GSI + daratumumab control group at the reviewer’s request. The purpose of the experiments in Figure 7 was to test iDuo-MM CAR-NK cell antitumor activity in vivo with the various treatment conditions and compare this to the antitumor activity of CAR-T cells.

Reviewer #3:

1. Most likely by error, the legend of Figure 4F mentions in addition to backbone iNK and iDuo-MM CAR-NK also serial restimulation of expanded PBNK, without such data shown in the Figure or mentioned in the results.

Yes. This was an error in the figure legend. This has been fixed. Thank you for noticing it.

2. For enhanced clarity, the experiments shown in Figure 7, panels A and B versus panels C to F should be described separately in the legend. For A and B, it should be indicated that NSG mice were first inoculated with MM.1S cells before investigating BCMA surface expression upon treatment with LY 3039478. The number of NSG mice mentioned (38) appears to refer to the separate adoptive therapy experiments shown in C to F, not A to B.

The reviewer makes a good point about clarity in this figure legend. We have rewritten this figure legend to address the comments from Reviewer 2.

REVIEWER COMMENTS

Reviewer #2 (Remarks to the Author):

I appreciate the newly provided data, which clarifies some of the issues but not all of them. First of all, I would like to emphasize that I appreciate the anti-myeloma activity of the final product iDuo-MM CAR-NK cells when they are combined with Daratumumab. The figure 5B shows that an anti-tumor effect size of >80% can be achieved in this combination, which is quite impressive.

But at the same time, it is important, and I think the reader who is interested in developing such products also will appreciate a lot- to define what the contribution of each modification is to the final effect size.

This was exactly the reason why I asked a number of questions in my previous comments. Now, with the newly provided data in figure 5B, I see that a large proportion of the anti-MM effect (possibly around 60-70%, if not more) is established by Daratumumab combined with iNK cells having all the modifications but lacking the BCMA-CAR. Thus these data also indicates that the BCMA-CAR contributes moderately to the whole anti-tumor effect. This is in agreement with the original data (data still present in figure 5B) that a maximum of 30-40% effect size can be achieved at day 30 after injection of a single dose of iDuo-MM CAR-NK cells only. Thus, one should conclude that the effect achieved by iDuo-MM CAR-NK cells alone is at most an intermediate effect if not a "modest" one. This intermediate effect seems, by the way, mainly due to the moderate activity of the used BCMA-CAR, because this BCMA CAR is also not generating a very powerful effect even when it is expressed on T cells (figure 6). Expectedly much higher anti-MM efficacies could be obtained if other CARs (with higher affinities) are used in these cells. I think the paper will certainly benefit from a discussion on the BCMA-CAR issue.

Having said that, the other important point which needs a thorough discussion in this manuscript is the possible contribution of IL5/IL5R gene to a) the in vivo persistence and b) to the whole anti-tumor effect. Here I have serious objections to the way how the authors present controls and interpret the data. Especially, I am disappointed to see that the authors are still referring to the data shown in figure 5F in which the numbers of iDuo-MM CAR-NK cells (and also the numbers of iNK cells) are compared to that of PBNK cells to draw the conclusion that this increased numbers of the iDuo-MM CAR-NK cells is caused by the expression of IL-15/IL5R fusion gene. Frankly speaking, I find it really odd to compare PBNK cells with iNK cells, especially if the aim is to show the specific contribution of IL-15/IL-15R fusion molecule to in vivo longevity and persistence of cells. PBNK cells are generated in a completely different way than iNK cells. Therefore their in vivo persistence can be inherently different than that of iNK cells regardless of the expression of IL-15/IL-15R fusion molecule. The correct control here is the iNKT cells having all modifications, except the modification for IL-15/IL-15R fusion molecule. These control cells should also be used to determine the contribution of IL-15/IL-15R fusion molecule to the overall clinical anti-tumor effect of the final product. This crucial control is missing in the whole manuscript. Thus this manuscript describes a working product but can not show us whether all these elements are necessary for the effectivity of this product, especially for the IL-15/IL-15R fusion molecule. This issue is important because there is currently much discussion on how crucial in vivo persistence is (and for how long) for the efficacy of CAR-NK cells (and also for CAR-T cells). An initial strong hit with relatively short persistency, for instance in case of Ciltacel, might circumvent the need for long term persistence. This could also be the case for CAR-iNK cells when they are combined with daratumumab, but unfortunately this manuscript lacks information on this important issue.

Finally, I appreciate the data provided for my other questions.

Reviewer #3 (Remarks to the Author):

The authors have added new experiments and more careful interpretation of the data to the newly revised version of the manuscript, further enhancing the significance of their study.

My remaining minor comments to the previous version were adequately addressed.

Please correct the following typing error: Duplication of "iDuo-MM CAR" in line 284.

Point-by-point response: NCOMMS-22-07254B “Multi-Antigen Targeting, Off-the-Shelf CAR-NK cells Demonstrate Durable Efficacy in Multiple Myeloma”

Please see our response (blue text) to the remaining critiques below:

Reviewer 2:

I appreciate the newly provided data, which clarifies some of the issues but not all of them. First of all, I would like to emphasize that I appreciate the anti-myeloma activity of the final product iDuo-MM CAR-NK cells when they are combined with Daratumumab. The figure 5B shows that an anti-tumor effect size of >80% can be achieved in this combination, which is quite impressive. But at the same time, it is important, and I think the reader who is interested in developing such products also will appreciate a lot to define what the contribution of each modification is to the final effect size. This was exactly the reason why I asked a number of questions in my previous comments. Now, with the newly provided data in figure 5B, I see that a large proportion of the anti-MM effect (possibly around 60-70%, if not more) is established by Daratumumab combined with iNK cells having all the modifications but lacking the BCMA-CAR. Thus, these data also indicates that the BCMA-CAR contributes moderately to the whole anti-tumor effect. This is in agreement with the original data (data still present in figure 5B) that a maximum of 30-40% effect size can be achieved at day 30 after injection of a single dose of iDuo-MM CAR-NK cells only. Thus, one should conclude that the effect achieved by iDuo-MM CAR-NK cells alone is at most an intermediate effect if not a “modest” one. The intermediate effect seems, but the way, mainly due to the moderate activity of the used BCMA-CAR, because this BCMA CAR is also not generating a very powerful effect even when it is expressed on T cells (Figure 6). Expectedly much higher anti-MM efficacies could be obtained if other CARs (with higher affinities) are used in these cells. I think the paper will certainly benefit from a discussion on the BCMA-CAR issue. Having said that, the other important point which needs a thorough discussion in this manuscript is the possible contribution of IL-15/IL-15R gene to a) the in vivo persistence and b) to the whole anti-tumor effect. Here I have serious objections to the way the authors present controls and interpret the data. Especially, I am disappointed to see that the authors are still referring to the data shown in figure 5F in which the numbers of iDuo-MM CAR-NK cells (and also the numbers of iNK cells) are compared to that of PBNK cells to draw the conclusion that the increased numbers of iDuo-MM CAR-NK cells is caused by the expression of IL-15/IL-15R fusion molecule. The correct control here is the iNK cells having all modifications except the modification for IL-15/IL-15R fusion molecule. These control cells should also be used to determine the contribution of IL-15/IL-15RF fusion molecule to the overall clinical anti-tumor effect of the final product. This crucial control is missing in the whole manuscript. Thus, this manuscript describes a working product but can not show us whether all these elements are necessary for the effectivity of this product, especially for the IL-15/IL-15R fusion molecule. This issue is important because there is currently much discussion on how crucial in vivo persistence is (and for how long) for the efficacy of CAR-NK cells (and also for CAR-T cells). An initial strong hit with relatively short persistence, for instance in case of Ciltacel, might circumvent the need for long term persistence. This could also be the case for CAR-iNK cells when they are combined with daratumumab, but unfortunately this manuscript lacks information on this important issue. Finally, I appreciate the data provided for my other questions.

We appreciate the reviewer's attention to detail and comments regarding the anti-BCMA CAR. The main thrust of the reviewer's critique seems to be the comparison in Figure 5B between the antitumor activity of backbone iNK cells (all modifications except for the anti-BCMA CAR) with daratumumab and iDuo-MM CAR-NK cells with daratumumab. Because of potency of daratumumab in this model, relatively modest differences are seen between these two groups in the in vivo experiments. However, we still believe that the CAR activity is generally robust. In the same figure, in the absence of daratumumab, iDuo-MM CAR-NK cells (all modifications including the CAR) exhibit significantly better tumor control relative to backbone iNK cells. This is also evident in Figure 5E. Significant activity of the anti-BCMA CAR in vitro is also shown in Figure 3A.

The second critique from the reviewer centers on comparisons between expanded PBNK cells and iDuo-MM CAR-NK cells in Figure 5. The reviewer is now asking for experiments to be performed with iNK cells that have all genetic modifications except IL-15RF to demonstrate the importance of this gene edit. The reviewer criticized PBNK cells as a control because iNK cells and PBNK cells are generated in different ways and could have inherent differences in persistence due to this fact. During the development of iDuo-MM CAR-NK cells, we did not generate an iPSC line with all gene edits except for IL-15RF. Doing so now would take approximately 6 months, and we would need another two months to set up and perform new in vivo experiments. Thus, it could not be done in a timely manner for revision of this manuscript. However, to address the reviewer's critique to the best of our ability and demonstrate the lack of significant antitumor activity and in vivo persistence in the absence of IL-15RF, we included additional in vivo data with iNK cells that had only one gene edit (*CD38* knockout) and no IL-15RF. Mice were engrafted with MM.1S cells and then treated with 3 doses of *CD38* knockout iNK cells with and without daratumumab. Tumor burden and iNK cell counts were monitored over time. Similar to other experiments, daratumumab had a moderate impact on tumor growth. However, *CD38* knockout iNK cells without any cytokine support were unable to control tumor. Their numbers in the peripheral blood were also very low (similar to those observed for PBNK cells in Figure 5). This data has been added as new Supplementary Figure 3 and shows that iNK cells, like PBNK cells, fail to persist and function in the absence of cytokine support.

Reviewer 3:

Please correct the following typing error: Duplication of "iDuo-MM CAR" in line 284.

This has been corrected.